# Microwave radiometry experiment for snow in Altay China: time series of *in situ* data for electromagnetic and physical features of snow pack

Liyun Dai[1], Tao Che[1,2]*, Yang Zhang[1], Zhiguo Ren[1,3], Junlei Tan[1], Meerzhan Akynbekkyzy[1], Lin Xiao[1], Shengnan Zhou[1], Yuna Yan[3], Yan Liu[4], Hongyi Li[1], Lifu Wang[5]

[1]Key Laboratory of Remote Sensing of Gansu Province, Heihe Remote Sensing Experimental Research Station, Northwest Institute of Eco-Environment and Resources, Chinese Academy of Sciences, Lanzhou, 730000, China.

[2]Center for Excellence in Tibetan Plateau Earth Sciences, Chinese Academy of Sciences, Beijing, 100101, China.

[3]University of Chinese Academy of Sciences, Beijing, 1000101, China.

[4] Institute of Desert Meteorology, China Meteorological Administration, Urumqi, 830002, China

[5]Altay National Reference Meteorological station, China Meteorological Administration, Altay, 836500, China.

*Correspondence to*: Tao Che (chetao@lzb.ac.cn)

**Abstract.** Snow depth is a key parameter in climatic and hydrological systems. Passive microwave remote sensing, snow process model and data assimilation are the main methods to estimate snow depth in large scale. The estimation accuracies strongly depend on input of snow parameters or characteristics. Because the evolving processes of snow parameters vary spatiotemporally, and are difficult to accurately simulate or observe, large uncertainties and inconsistence exist among existing snow depth products. Therefore, a comprehensive experiment is needed to understand the evolution processes of snow characteristics and their influence on microwave radiation of snowpack, to evaluate and improve the snow depth and SWE retrieval and simulation methods. An Integrated Microwave Radiometry Campaign for snow (IMCS) was conducted at the Altay National Reference Meteorological station (ANRMS) in Xinjiang, China, during snow season of 2015/2016. The campaign hosted a dual polarized microwave radiometer operating at L, K and Ka bands to provide minutely passive microwave observations of snow cover at a fixed site, daily manual snow pit measurements, ten-minute automatic 4-component radiation and layered snow temperatures, covering a full snow season of 2015/2016. The measurements of meteorological and underlying soil parameters were requested from the ANRMS. This study provides a summary of the obtained data, detailing measurement protocols for microwave radiometry, in situ snow pit and station observation data. A brief analysis of the microwave signatures against snow parameters is presented. A consolidated dataset of observations, comprising the ground passive microwave brightness temperatures, in situ snow characteristics, 4-component radiation and weather parameters, was achieved at the National Tibetan Plateau Data Center, China. The dataset is unique in providing continuous daily snow pits data and coincident microwave brightness temperatures, radiation and meteorological data, at a fixed site over a full season. The dataset is expected to serve the evaluation and development of microwave radiative transfer models and snow process models. The consolidated data are available at

http://data.tpdc.ac.cn/zh-hans/data/df1b5edb-daf7-421f-b326-cdb278547eb5/ (doi: 10.11888/Snow.tpdc.270886) (Dai, 2020).

**Key words:** Snow, Microwave radiometry, Snow pit, Experiment

## 1 Introduction

Seasonal snow cover plays a critical role in climate and hydrological systems (Cohen, 1994; Ding et al., 2020; Barnett et al., 2005; Immerzeel et al., 2010) by its high albedo, thermal insulation, fresh water reserves and its phase change processes. Snow cover can be accurately identified by optical remote sensing. However, the snow surface albedo is controlled by snow characteristics (Aoki et al., 2003 and 2000), and variations in snow characteristics cause uncertainties in albedo estimation. Snow depth and snow water equivalent (SWE) are currently estimated using passive microwave at global and regional scales (Pullianen et al., 2020; Tedesco and Narvekar, 2010; Jiang et al., 2014; Che et al., 2008). Although several global and regional snow depth and SWE products have been released, large uncertainties exist in these products because of the spatio-temporal variations in snow characteristics (Xiao et al., 2020; Mortimer et al., 2020; Che et al., 2016; Dai et al., 2012; Dai and Che, 2022). Therefore, the observation of electromagnetic and physical parameters of snowpack are necessary to improve understanding of the electromagnetic radiation process of snowpack to enhance the estimation accuracy of snow surface albedo and snow depth.

To evaluate and improve snow depth and SWE retrieval methods from passive microwave remote sensing observations and to combine remote sensing technologies with modeling and data assimilation methods to produce the most accurate products, a few large or systematic experiments or campaigns have been conducted on electromagnetic and physical characteristics measurement of snow cover. These experiments are summarized in table 1. The Cold Land Processes Field Experiment (CLPX) (https://nsidc.org/data/clpx/index.html) , one of the most well-known experiments, was carried out from winter of 2002 to spring of 2003 in Colorado, USA (Cline et al., 2003). During the campaign, snow pits were collected in February and March of 2002 and 2003 to coincide with airborne and ground remote sensing observations. NASA SnowEx campaign (https://nsidc.org/data/snowex) was conducted in 2017 in Colorado to test and develop algorithms for measurement of SWE in forested and non-forested areas by providing multi-sensor observations of seasonally snow-covered landscapes (Brucker et al., 2017). The campaign is still ongoing and will be conducted in other areas with different snow conditions. In northern Canada, mobile sled-mounted microwave radiometers were deployed in forest, open and lake environments from November 2009 to April 2010 and snow characteristics within the footprints of radiometers were measured to improve understanding the influence of snow characteristics on brightness temperatures (Derksen et al., 2012; Roy et al., 2013). These microwave experiments were of mobile observation. In these experiments, there were multiple observation sites for different land cover, but relative short temporal range. The snow pit observations were used to evaluate snow microwave emission model in different land cover (Tedesco and Kim, 2006; Royer et al., 2017), but they did not exhibit the evolution of snow parameters.

In the Arctic region, the Nordic Snow Radar Experiment (NoSREx) campaign was conducted at a fixed field in Sodankylä, Finland, during 2009 ~ 2013 (Lemmetyinen et al., 2016). This experiment provided a continuous time series of active and passive microwave observations of snow cover at a representative location of the Arctic boreal forest area spanning an entire winter season and matched

snow pit observations were made weekly. In Asia, snow pit work at 3 or 4 day intervals was conducted simultaneously with radiation budget observations during winter of 1999/2000 and 2000/2001 to analyze the effects of snow physical parameters on albedo (Aoki et al., 2003). The NoSREx and Japan radiation experiments were fixed field observation, which provided longer time series of data than CLPX and SnowEx. These experiments were conducted in deep snow areas, and the weekly observation could reflect general evolution process of snow characteristics but might miss some key details that occur at sub-weekly scales. In the Tibetan plateau with shallow snow cover, multiple years of microwave radiometry observation at L band were conducted to study passive microwave remote sensing of frozen soil (Zheng et al., 2019, 2021a and 2021b). However, in the long term series of experiment, no snow pit was measured and the microwave radiometry observation was performed at L band which is insensitive to snowpack.

**Table 1 Summary of existing experiments for microwave and optical radiation and physical features of snowpack**

| Campaign | Location | Temporal range | Observation content |
|---|---|---|---|
| CLPX | Different sites in Colorado, | February and March of 2002 and 2003 | Inconsecutive multiple sensor observation, including microwave radiometry over snow, and matched snow pit measurements were conducted at different sites with short temporal range. |
| SnowEx-year 1 | Grand Mesa, and Senator Beck Basin, Colorado | February of 2017 | Inconsecutive multiple sensor observation, including microwave radiometry over snow, and matched snow pit measurements were conducted at different sites with short temporal range. |
| CMRES[1] | Mobile observation at Forest, open and lake in the northern Canadian region | November of 2009-April of 2010 | Mobile microwave radiometry and snow pit observation within footprint of radiometer. Short temporal range and inconsecutive observation |
| NoSREx | Fixed site in Sodankylä, Finland | Snow season during 2009-2013 | Consecutive microwave radiometry and SAR observation over snow, and weekly snow pit measurement |
| JERBS[2] | Fixed site in Japan | Snow season during 1999-2000 | Consecutive optical radiation observation over snow and consecutive snow pit measurement at 3 or 4-day interval. |
| IMCS | Fixed site in China | November of 2015-March of 2016 | Consecutive microwave radiometry and optical radiation observation, and consecutive daily snow pit measurements. |

Note: [1]CMRES: Microwave radiometry experiment on snow cover conducted in northern Canada

[2]JERBS: Experiment of radiation budget over snow cover in Japan

To understand the evolution of snow characteristics and their influence on passive microwave brightness temperatures and radiation budget, an integrated experiment on snow was conducted during a

full snow season, in Altay, China. The experiment was designed to cover periods from snow-free conditions to eventual snow melt-off during 2015/2016. The microwave radiometry measurements at L, K and Ka bands for multiple angles were complemented by a dual-polarized microwave radiometer with 4-component radiation and daily in situ observations of snow, soil and atmospheric properties, using both manual and automated methods. The data of electromagnetic and physical parameters were further consolidated and organized to be easily read and utilized.

The dataset is unique in providing continuous daily snow pits data over a snow season at a fixed site and matched microwave brightness temperatures, radiation and meteorological data. In the next section, the experiment location, parameters, and parameter measurement protocols are described; section 3 introduces the consolidated data which was released at the National Tibetan Plateau Data Center, China; section 4 presents content of brightness temperature, 4-component radiation, snow pit data, soil temperature and moisture, and meteorological data; section 5 discusses the possible applications and uncertainties; and finally the conclusions are summarized in section 6.

## 2 Description of experiment setup

### 2.1 Measurement location

The Integrated Microwave Radiometry Campaign for snow (IMCS) was performed during the 2015/2016 snow season (from November 27, 2015 to March 25, 2016) at the Altay National Reference Meteorological station (ANRMS) (N47°44'26.58", E 88°4'21.55") which is approximately 6 km from the foot of Altay mountain in the northwest China (Figure 1). Altay mountain with elevation up to 3000 m, running northwest and southeast, is at the junction of China, Russia, Mongolia and Kazakhstan, and provides snow water resources for these four countries. The average annual maximum snow depth measured in this station is approximately 40 cm, with a maximum over 70 cm. In the southwest of Altay mountain, crop land and desert with flat terrain are the dominant land covers. Snow cover is critical fresh water for the irrigation in this area. In this experiment, measurements included microwave radiometry, 4-component radiation, snow pit and soil parameters. The test site of this experiment was four neighboring bare rectangle fields in the ANRMS with areas of 2500m$^2$ (black rectangle filed in Figure 1), 2500m$^2$ (pink rectangle field in Figure 1), 200m$^2$ (red rectangle field in Figure 1) and 400 m$^2$ (blue rectangle field in Figure 1), respectively.

In the pink field, the ground-based microwave radiometer was set up in the middle of the field, facing south to collect brightness temperatures of snow cover. The black field behind the microwave radiometers (north of the radiometers) was for manual snow pit data collection. The microwave radiometer observations and snow pit data collection were conducted by Northwest Institute of Eco-Environment and Resources, Chinese Academy of Science (NIEER) from November 27, 2015 to March 25, 2016 (After March 25, 2016, snow melted out).

The blue field was for meteorological measurements including wind speed, wind direction, air temperature, air wetness, air pressure, precipitation, soil temperature, soil moisture among others. These parameters were automatically obtained from instruments, and the instruments setup and data collection were operated by ANRMS. This station also has daily manual observation of snow depth and SWE. In this experiment, we requested the wind, air pressure, air wetness, air pressure, soil temperature and moisture data during this experiment from ANRMS. The red field was designed for automatic

measurement of layered snow temperatures, snow density, SWE, snow depth, and albedo. These automatic measurement instruments were installed and maintained by NIEER, and started working from 2013. However, during the experiment, the instruments for snow density and SWE did not work, and we only collected layered snow temperatures and 4-component radiation.

Because the four observation fields are located within the domain of the station and the distance between them are less than 100m, the snow characteristics and soil and weather conditions are thought to be the same. Overall, the experiment performed a systematic observation covering electromagnetic and physical features of snow pack, providing data for studies on snow remote sensing and models.

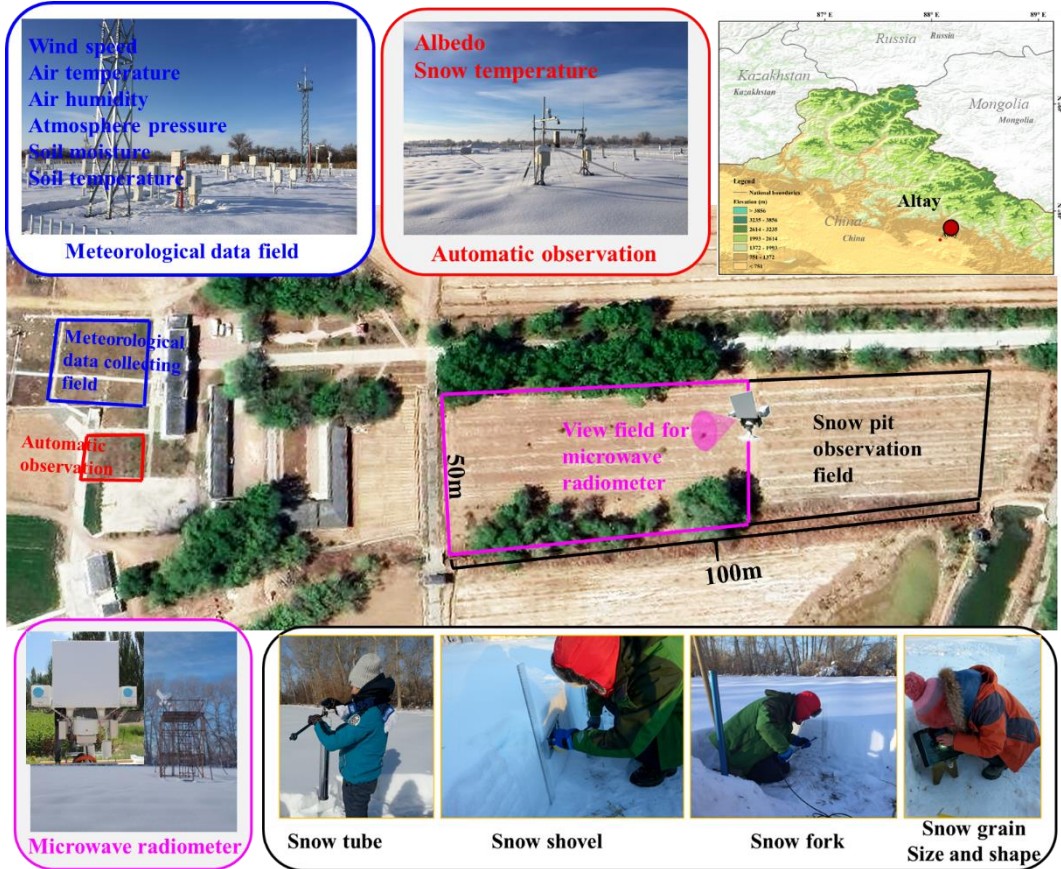

**Figure 1: Location of the Altay National Reference Meteorological station (ANRMS) in Asia and the distribution of three experiment fields in the ANRMS. The black rectangle represents the field used for snow characteristics (approximately 40 m × 50 m) including snow layering, layer thickness, snow density, snow grain size and shape of each layer, and microwave radiometers (approximately 60 m × 50 m) observations. The blue rectangle is the field for meteorological and soil data collection operated by the ANRMS. The red rectangle field is for the automatically observation of the snow temperature, and 4-component radiation, designed by Northwest Institute of Eco-Environment and Resources, Chinese Academy of Science (NIEER).**

## 2.2 Measurement methods

The microwave signatures from snowpack vary with snow characteristics, soil and weather conditions. In this experiment, the measurements include microwave radiometry observation to collect brightness temperature, manual snow pit observation to collect snow physical parameters, automatic observation to collect 4-component radiation and snow temperatures, and meteorological observation

which contains weather data and soil data.
**2.2.1. Microwave radiometry**
The brightness temperatures at 1.4, 18.6, 36.5 GHz for both polarization (Tb1h, Tb1v, Tb18h, Tb18v,
Tb36h, Tb36v) were automatically collected using a six-channel dual polarized microwave radiometer
RPG-6CH-DP (Radiometer Physics GmbH, Germany,
https://www.radiometerphysics.de/products/microwave-remote-sensing-instruments/radiometers/). The
technical specifications of the RPG-6CH-DP are described in Table 2. The RPG-6CH-DP contains a
built-in temperature sensor which can measure air temperature. The automated data collection frequency
was set to 1 minute.

**Table 2. Technical Specifications of the RPG-6CH -DP Microwave Radiometer.**

| Parameter | Value |
|---|---|
| Manufacturer | Radiometer Physics GmbH |
| System noise temperatures | <900 K |
| Bandwidth | 400MHz (20MHz for 1.4 GHz) |
| System stability | 0.5 K |
| Dynamic range | 0~400 K |
| Frequencies (GHz) | 1.4, 18.7, 36.5 |
| Polarizations | V, H |
| Internal calibration | Internal Dicke switch and software control for automatic sky tilt calibration |
| Receiver and antenna thermal stabilization | < 0.015 K |
| Antenna sidelobe level | < -30 dBc |
| Optical resolution (HPBW) | 6.1º (11º for 1.4 GHz) |
| Incidence angle | 0~90º |
| Azimuth angle | 360º |


Before the snow season, a platform with height of 5 m, length of 4 m and width of 2 m was
constructed in the experiment field (Figure 2). A 4-m orbit was fixed on the platform. The RPG-6CH-DP
was set up on the orbit and could be moved along the orbit. The microwave radiometers at K and Ka
bands began working from November 27, 2015, but the L band radiometer did not work until January 30,
2016.These radiometers were sky tipping calibrated, and the calibration accuracy is 1 K. In clear sky
conditions, the sky brightness temperatures were approximately $29.7\pm0.3$ K at 18.7 GHz for both
polarizations and $29.3\pm0.9$ K at 36.5 GHz for both polarizations. But the sky brightness temperature at
L band showed large fluctuation. They ranged from -1 to 8 K for horizontal polarization, and 1 to 16 K
for vertical polarization.
Generally, the radiometers were fixed in the middle of the orbit to observe snow cover with incidence
angle of 50º. Multi-angle observations were conducted after every big snowfall, and every 5 days in the
stable period. In the melt period, observation frequency increased. There are total seventeen multi-angle
observation (December 3, 19, and 30; January 3, 8, 13, 18, 3, and 28; February 3; March 3, 10, 15, 22,
26, 28, and 31) when the radiometer was set to scan the ground at different incidence angles at two ends
of the orbit and the middle place of the orbit. Although the view fields of the antennas for 1.4 GHz, 18
GHz and 36 GHz did not completely overlap, the measured results showed that the brightness
temperatures observed by radiometers at the left, middle and right of the orbit varied less than 1 K.

Therefore, the snow and soil characteristics were considered homogeneous within the view fields of the three antennas.

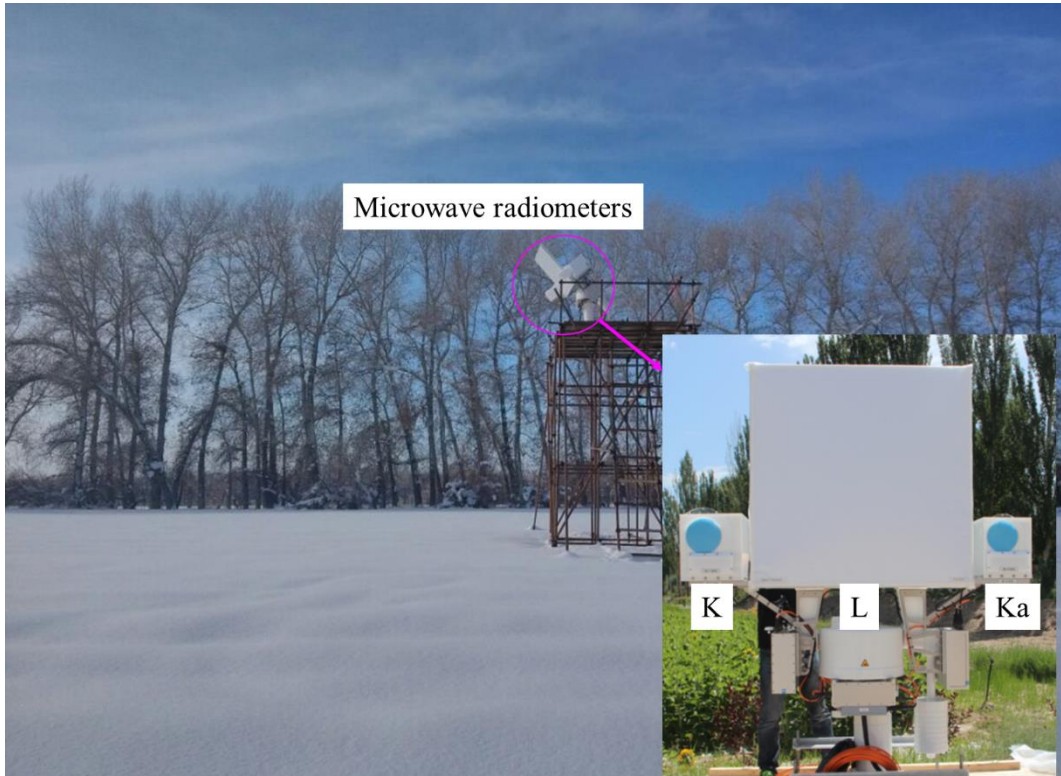

**Figure 2 Ground-based microwave radiometer observation.**

### 2.2.2 Snow pit measurement

The snow characteristics were obtained by manual snow pit measurements in the black field, including snow layering, snow layer thickness, grain size, snow density, and snow temperatures. These data were daily collected during 8:00-10:00 am local time, from November 27, 2015 to March 25, 2016, except 7 days (please see Table 3). Although the snow temperatures were manually measured at snow pits, the automatically collected snow temperatures in the red field were utilized in this study, because the temperature measured at snow pits could not reflect the natural temperature profile when the snow pits exposed to air.

**Table 3. Variables collected by manual daily snow pit measurement in black field in figure 1, and their observation instruments, observation time and frequencies.**

| Parameter | Instruments | Precision | Layering style | Observation time or frequency | Absent date |
|---|---|---|---|---|---|
| Layer thickness (cm) | Ruler | 0.1cm | Natural layering | | no |
| Snow density $(g/cm^3)$ | Snow tube (Chinese Meteorological Administration) | pressure:0. 1$g/cm^2$, snow depth: 0.1 cm | Whole snowpack | local time 8:00-10:00 am | no |
| Snow density $(g/cm^3)$ | Snow shovel (NIEER) | weight: 0.01g, volume: 1$cm^3$ | Every 10 cm | | January 2-3, 2016; |

| | | | | February 20, 2016 |
|---|---|---|---|---|
| Snow density (g/cm$^3$) and | Snow fork (Toikka Enginnering Ltd.) | 0.0001g/cm$^3$ | Every 5 cm | |
| Liquid water content (%) | Snow fork | 0.001% | Every 5 cm | |
| Snow grain size (mm) | Anyty V500IR/UV | 0.001mm | Natural layering | December 24, 31, 2015; |
| Snow grain shape | Shape card | N/A | Natural layering | January 1-3, 23, 2016, February 20, 2016 |

The first step of snow pit measurement is making a snow pit. In the black field, a new snow pit was dug each day. A spade was used to excavate snow pit. The length of the snow pit profile was approximately 2m to make sure all parameters were measured from unbroken snowpack. The width of the snow pit was approximately 1m. The snow pit section was made as flat as possible using a flat shovel or ruler. When the snow profile is exposed to air for a long time, the snow characteristics will be influenced by environment and will be different from the natural snow characteristics. In order to make sure every observation conducted on natural snow pit, the snow pit was backfilled with the shoveled snow after finishing all observations, and the new snow pit in the following day was made at least 1-m distance from the last snow pit. After finishing a snow pit, the natural snowpack stratification was then visually determined, and the thickness of each layer was measured using a ruler.

The third step was measuring grain size and shape type in each layer. The grain size and type within each natural layer were estimated visually from a microscope with an "Anyty V500IR/UV" camera (Figure 3a). A software "VIEWTER Plus" matched the microscope was used to measure grain size. The grain type was determined based on Fierz et al. (2009). In this experiment, we utilized the length of longest axes and the length of shortest axes to describe grain size (Figure 3b). When using the software to measure the grain size, a reference must be needed. In this experiment, a ruler with 0.5 mm marking was used as a reference (Figure 3c). We adjusted the focus of the camera to make sure the grains at the clearest status in camera to take photos, and the photo of ruler scale was taken at the same focus. If the thickness of one layer was less than 10 cm, measurements were performed at the top and bottom of the layer. If the thickness was greater than 10 cm, measurements were performed at the top, middle, and bottom of the layer. For each layer, at least 5 photos were taken, and at least 10 typical grains were chosen to measure the longest axes length and the shortest axes length in the photos of each layer. Each layer had at least 10 groups of longest and shortest axes length; the final grain size was the average of these values. Figure A1 presents an example of the original photos of grains in each layer, and Table A1 shows the matched record of longest and shortest axis length.

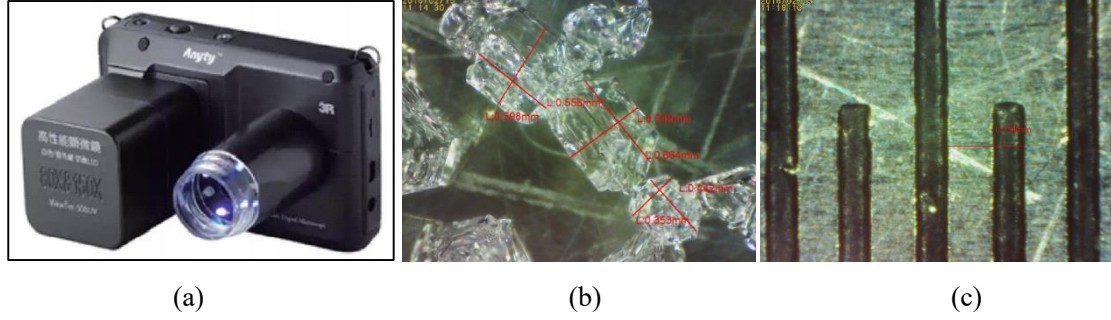

|  (a) | (b) | (c) |

**Figure 3: Picture of microscope "Anyty V500IR/UV" (a), the measured longest axes lengths and shorteast axes length of particles (b), and the reference ruler scale (c).**

Snow density was measured using three instruments: snow tube, snow shovel and Snow Fork (Figure 4). The snow tube instrument, designed by Chinese Meteorological administration, contains a metal tube with the base area of 100 cm$^2$ and the length of 60 cm, and a balance (figure 4a). It was utilized to measure the snow density of a whole snowpack by weighing the snow sample. The snow shovel is a 1500 cm$^3$ wedge-type sampler, and its length, width and height are 20 cm, 15 cm, and 10 cm, respectively (figure 4b). It was utilized to measure snow density every 10 cm (0-10 cm, 10-20 cm, 20-30 cm…). The Snow Fork is a microwave resonator that measures the complex dielectric constant of snow, and adopts a semi-empirical equation to estimate snow density and liquid water content based on the complex dielectric. The Snow Fork (figure 4c) was utilized to measure snow density and liquid water content at 5-cm intervals starting 5 cm above the snow/soil interface (5cm, 10cm, 15 cm, 20cm…). In order to decrease the observation error, every measurement was conducted three times. If there is an abnormal value, the fourth measurement would be performed to make sure the accuracy. Table A2 is an example record table for snow density. The average value of the three-time observation was the final value.

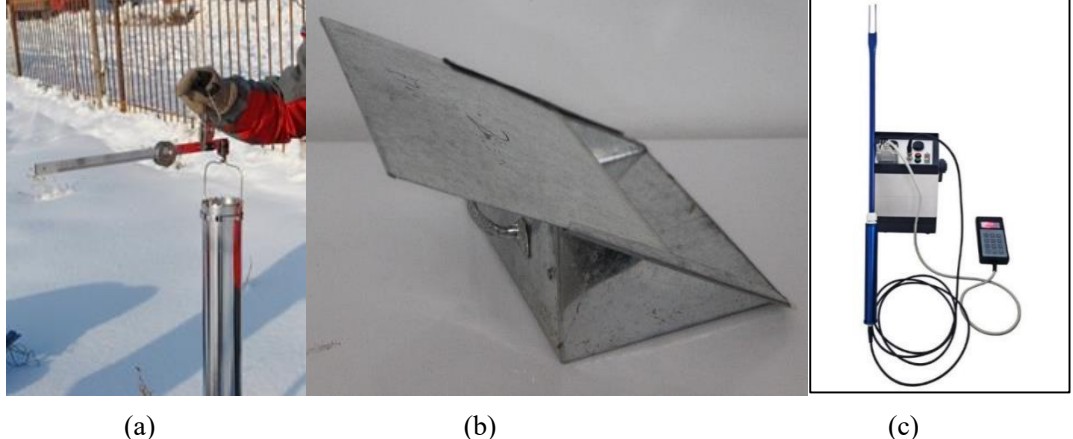

|  (a) | (b) | (c) |

**Figure 4: Three instruments for snow density: Snow tube (a), Snow shovel (b), and Snow Fork (c).**

### 2.2.3 Automatic radiation and temperature measurement

In the red field, the 4-component radiation was automatically measured by Component Net Radiometer (NR01) manufactured by Hukseflux, and layered snow temperatures was measured by Campbell 109S temperature sensors, respectively. The temperature sensors were set up on a vertical pole which was vertically inserted in the soil (Figure 5). The heights of the sensors are 0 cm, 5 cm, 10 cm, 15 cm, 25 cm, 35 cm, 45 cm, and 55 cm above soil/snow interface. The snow temperatures at these heights

were collected every ten minute.

The NR01 net radiometer was set up to measure the energy balance between incoming short-wave and long-wave far infrared radiation versus surface-reflected short-wave and outgoing long-wave radiation. The range of short wave is 285~3000nm, and the range of long wave is 4.5~40um. The 4-component radiation was automatically recorded every ten minutes. In addition, the sensor is equipped with a Pt100 temperature sensor for parallel recording of the sensor temperature.

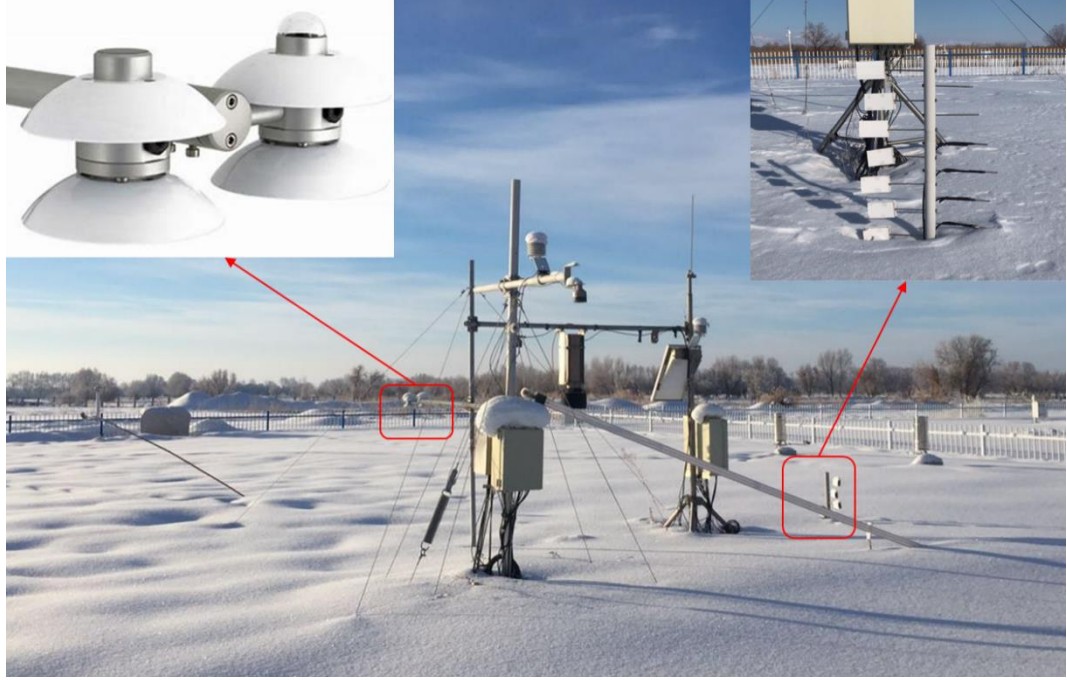

**Figure 5: Set up of temperature sensors and CNR01 in the red field.**

### 2.2.4 Meteorological observation

The meteorological data include air temperature, air pressure and humidity, wind speed, soil temperature at -5cm, -10 cm, -15cm and -20 cm and soil moisture at -10 cm and -20 cm. These parameters are routine observations conducted at ANRMS, and were obtained through request from ANRMS. The instruments used for soil and weather parameters observations are produced by China Huayun Meteorological Technology Group corporation. The measurement parameters and their measurement instruments are listed in table 4.

**Table 4. Automatically observed variables and the observation instruments, observation time and frequencies.**

| Parameter | Instruments | Precision | Layering style | Observation time or frequency |
|---|---|---|---|---|
| Snow temperature(°C) | Temperature sensors (Campbell 109S) | 0.001 °C | 0 cm, 5 cm, 10 cm, 15 cm, 25 cm, 35 cm, 45 cm, and 55 cm | Ten-minute |
| 4-component radiation (W/m²) | Component Net Radiometer NR01 (Hukseflux) | 0.001 W/m² | 6 feets above ground | Ten-minute |

| | | | | |
|---|---|---|---|---|
| Soil temperature (°C) | Soil temperature sensor (China Huayun) | 0.1 °C | -5cm, -10 cm, -15cm and -20 cm | Hourly |
| Soil moisture (%) | Soil moisture sensor (DZN3, China Huayun) | 0.1% | -10 cm and -20 cm | Hourly |
| Air temperature (°C) | Thermometer screen (China Huayun) | 0.1 °C | 6 feet above ground | Hourly |
| Air pressure (hPa) | Thermometer screen (China Huayun) | 0.1 hPa | 6 feet above ground | Hourly |
| Air humidity (%) | Thermometer screen(China Huayun) | 1% | 6 feet above ground | Hourly |
| Wind speed (m/s) | Wind sensor(China Huayun) | 0.1m/s | 10 m above ground | Hourly |


The air temperature, pressure and humidity were collected using temperature and wetness sensor in
thermometer screen, the wind speed and direction were measured using wind sensor set up at 10 m on a
tower. Soil moisture and temperature were automatically measured using moisture sensor and
temperature sensor. Figure 6 depicts the instruments for these observations.

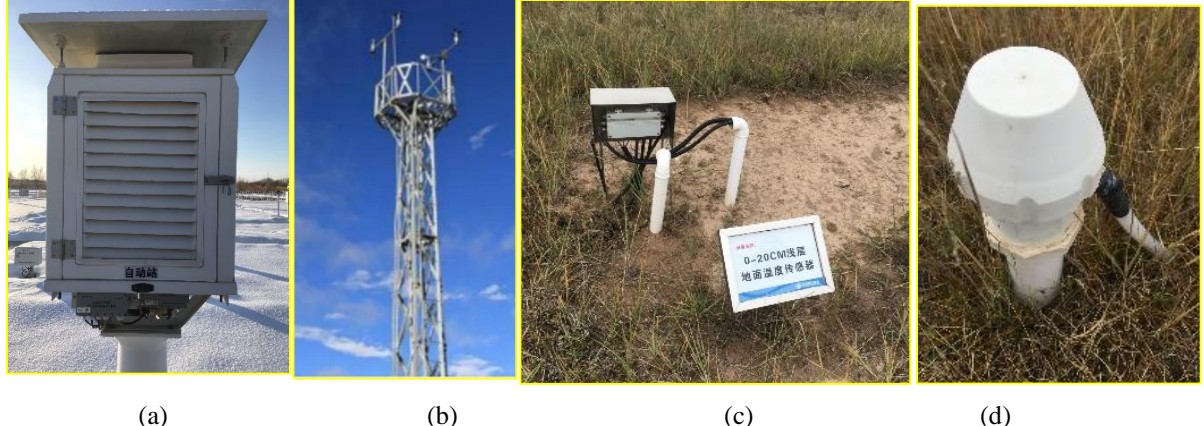

287       (a)                              (b)                              (c)                              (d)

**Figure 6: Instruments for observation of air temperature and wetness (a), wind speed (b), soil temperature**
**(c) and soil moisture (d).**
**3 Description of consolidated IMCS dataset**
The microwave brightness temperature, snow parameters, meteorological data were recorded in
different formats, and the observation frequencies and times were different. These data must be
reorganized and consolidated for ease of use. The values from the three-time measurements for snow
density in each layer were averaged to obtain the final snow density. The length of the longest and shortest
axes of particles in each photo were measured using the software. The average lengths of longest and
shortest axes from all photos in each layer were obtained as the final grain size. The daily snow pit data
were finally consolidated into a NetCDF file "snow pit data.nc".
The time series of automated layered snow temperature and 4-component radiation data were firstly
processed with removal of abnormal values and gap fill, and then were consolidated into a NetCDF file
"ten-minute 4 component radiation and snow temperature.nc". The ground-based brightness
temperatures and the formatted weather and soil data requested from ANRMS were provided 'as is'.
Brightness temperature data were divided into time series of brightness temperature and multi-angle
brightness temperatures, and separately stored in two NetCDF files, and the weather and soil data were
consolidated into a NetCDF file "hourly meteorological and soil data.nc". Table 3 describes the contents
of the provided dataset.
**1) Brightness temperatures data:**
1 Minutely brightness temperature at 1.4 GHz, 18 GHz and 36 GHz for both polarizations at incidence
angle of 50°. This data include date, time, incidence angle, azimuth angle, and brightness temperatures
at the three bands for both polarizations.
2 Seventeen groups of calibrated brightness temperature at 1.4 GHz, 18 GHz and 36 GHz for both
polarizations at different incidence angles (30, 35, 40, 45, 50, 55, 60°). This data include date, incidence
angles, azimuth angle, brightness temperatures at the three bands for both polarizations.
**2) Manual snow pit data:**
Daily snow pit data include date, snow depth, layered snow thickness, average longest axis, average
shortest axis, grain shapes of each layer; layered snow density using snow fork (snow density at different
heights, such as SF_5cm, SF_10cm, SF_15cm), snow density using snow tube, layered snow density
using snow shovel (such as SS_0-10cm, SS _10-20cm, SS _20-30cm, SS _30-40cm).
**3) Automated snow temperature and radiation data**
Ten-minute 4-component radiation and snow temperature data include date, time, short-wave incident
radiation, short-wave reflected radiation, long-wave infrared incident radiation, long-wave infrared
reflected radiation, sensor temperature, and snow temperatures at different heights (such as ST_0cm,
ST_5cm)
**4) Meteorological and soil data:**
Hourly weather data include date, hour, air temperature, pressure, humidity, wind speed, soil temperature
at 5 cm, 10 cm, 15 cm and 20 cm, and soil moisture at 10 cm and 20 cm.

**Table 3 Description of consolidated data**

| Data | Content | File name | Variables |
|---|---|---|---|
| Brightness temperature | Brightness temperature | TBdata.nc | Year, month, day, hour, minute, second, Tb1h, Tb1v, Tb18h, Tb18v, Tb36h, Tb36v, incidence angle, azimuth angle |
| | Multi-angle brightness temperatures | TBdata-multiangle.nc | Year, month, day, hour, minute, second, Tb1h, Tb1v, Tb18h, Tb18v, Tb36h, Tb36v, incidence angle, azimuth angle |
| Manual snow pit data | Layer thickness, layered grain size and shape, snow density | Daily snow pit data.nc | Year, month, day, snow depth, th1, Lg1, Sg1, th2, Lg2, Sg2, th3, Lg3, Sg3, th4, Lg4, Sg4, th5, Lg5, Sg5, th6, Lg6, Sg6, Stube, SS_0-10, SS_10-20, SS_20-30, SS_30-40, SS_40-50, SF_5, SF_10, SF_15, SF_20, SF_25, SF_30, SF_35, SF_40, SF_45, SF_50, shape1, shape2, shape3, shape4, shape5, shape5 |
| Automated snow temperature and radiation data | 4-component radiation, snow temperature | Ten-minute 4 component radiation and snow temperature.nc | Year, month, day, hour, minute, SR_DOWN, SR_UP, LR_DOWN, LR_UP, T_Sensor, ST_0cm, ST_5cm, ST_15cm, ST_25cm, ST_35cm, ST_45cm, ST_55cm |

| Meteorological and soil data | meteorological data, soil moisture and temperature | Hourly meteorological and soil data.nc | Year, month, day, hour, Tair, Wair, Pair, Win, SM_10cm, SM_20cm, Tsoil_5cm, Tsoil_10cm, Tsoil_15 cm, Tsoil_20cm |
|---|---|---|---|

Note: th: snow thickness, Lg: long axis, Sg: short axis, shape: grain shape;
Stube: snow density observed using snow tube, SS: snow density observed using snow shovel, SF: snow density
observed using snow fork; ST: snow temperature; SR_DOWN: downward short-wave radiation, SR_UP: upward
short-wave radiation, LR_DOWN, downward long-wave radiation, LR_UP: upward long-wave radiation, T_sensor:
sensor temperature; Tair: air temperature, Wair: air wetness, Pair: air pressure, Win: wind speed.

**4 Overview and preliminary analysis of collected data from IMCS**

**4.1 Snow characteristics**

**4.1.1 Layering grain size and grain shape**

During 2015/2016, snow cover began on 25 November of 2015, and ended on March 25 of 2016. During this snow season, there were seven snowfall events and each formed a distinct snow layer except for the third event whose layering became indistinguishable from the second layer (Figure 7 gray). The fourth event was the biggest, after which time snow depth started to decrease and snow density increased. Snow cover began melting on March 14 and snow depth declined to zero within 10 days.

Grain sizes within all layers increased during the snow season, except in the bottom layer where grain size experienced a decrease from December 28 to January 20 (Figure 8). In the vertical profile, grain size increased from top to bottom with the snow age. The grain size of the fresh snow was approximately 0.3 mm during the experiment. The biggest long and short axis were up to 6 cm and 4 cm, respectively, and occurred in Layer 1 in during the melt period. The length of short axes is approximately 0.7 of the length of long axes. The grain shape generally developed from rounded grains to facet crystals, and then to depth hoar. After March 13, 2016, the minimum air temperature increased to above 0°C, snowpack melt accelerated, and the grain shape developed to melted forms (Figure 7).

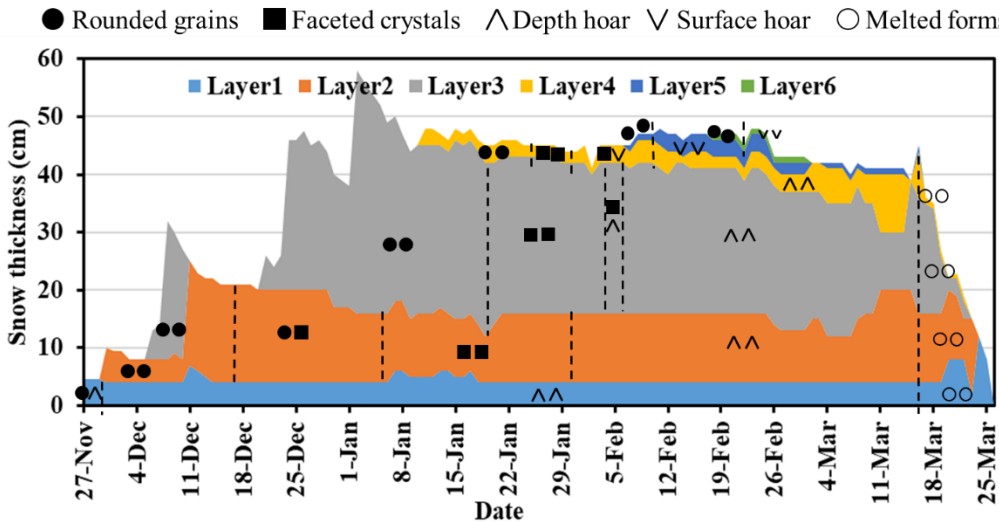

**Figure 7: Daily variation in snow layers and grain shape in each layer from November 27, 2015 to March 25,**

**2016.**

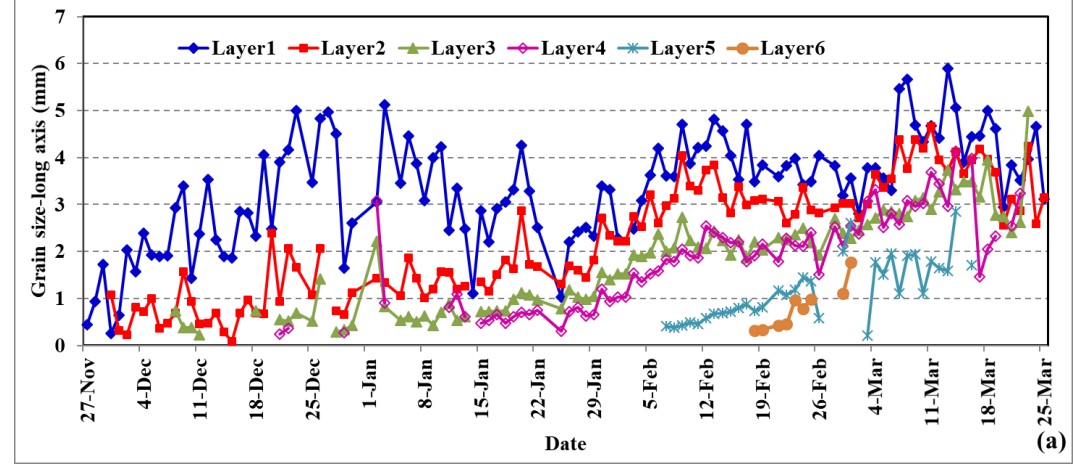


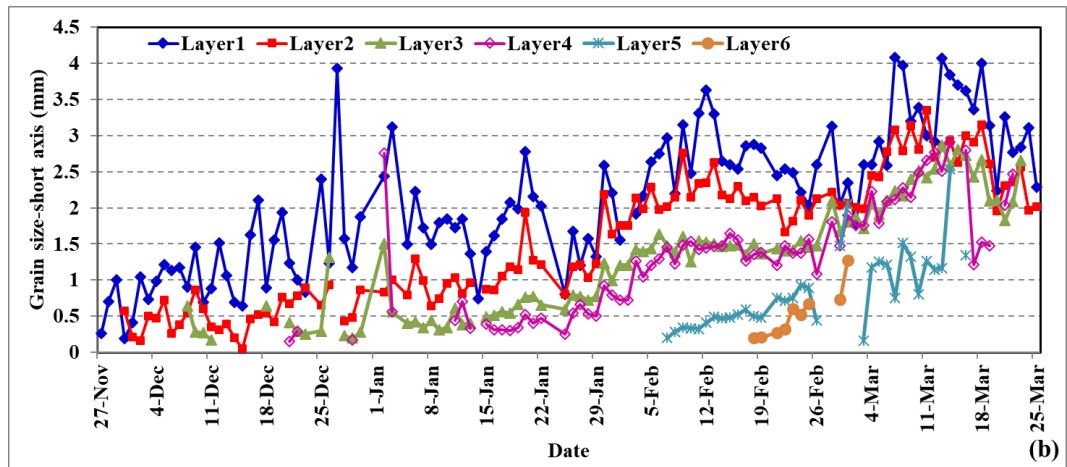


**Figure 8: Daily variation in grain size within each layer from November 27, 2015 to March 25, 2016. The thickness of each layer is presented in figure 9.**


**4.1.2 Snow density**

Snow densities measured by three different instruments shows that the density of fresh snow ranged between 0.05~1.0 g/cm$^3$ (Figure 9). The snow densities increased with snow age, and remained stable after reaching ~0.2-0.25g/cm$^3$. From March 14 on, snow densities abruptly increased, and the maximum value reached was over 0.45g/cm$^3$. In the vertical profile, snow density increased from top to bottom in the accumulation phase, but after January 3, 2016, snow densities in the middle layers were larger than those in the bottom and upper layers due to the well-developed depth hoar of bottom layer. In the melting phase, snow densities in all layers showed little difference. Snow fork provided most detail snow density profile, but it systematically underestimated snow density compared with snow tube and snow shovel by 24% (Dai et al., 2022).


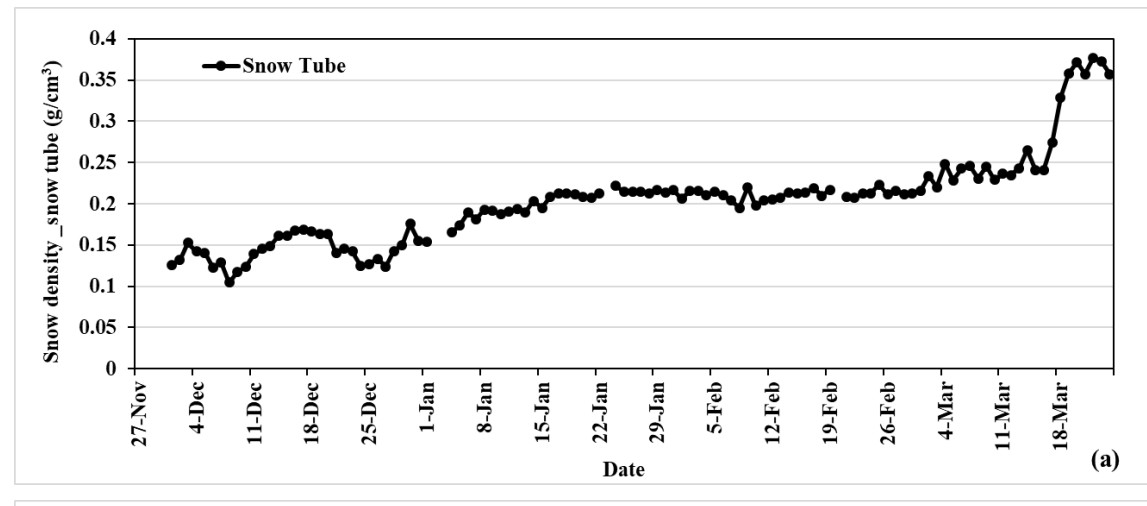


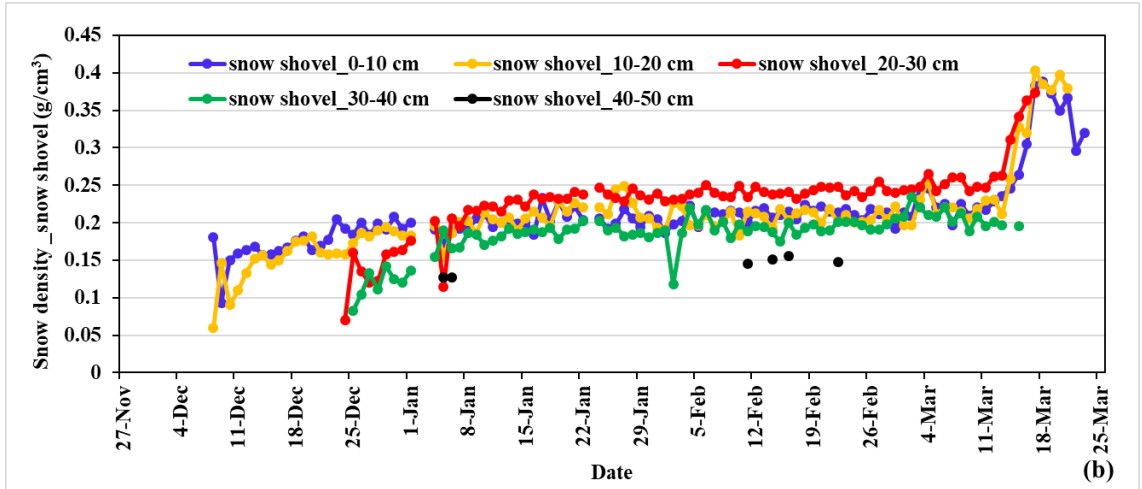


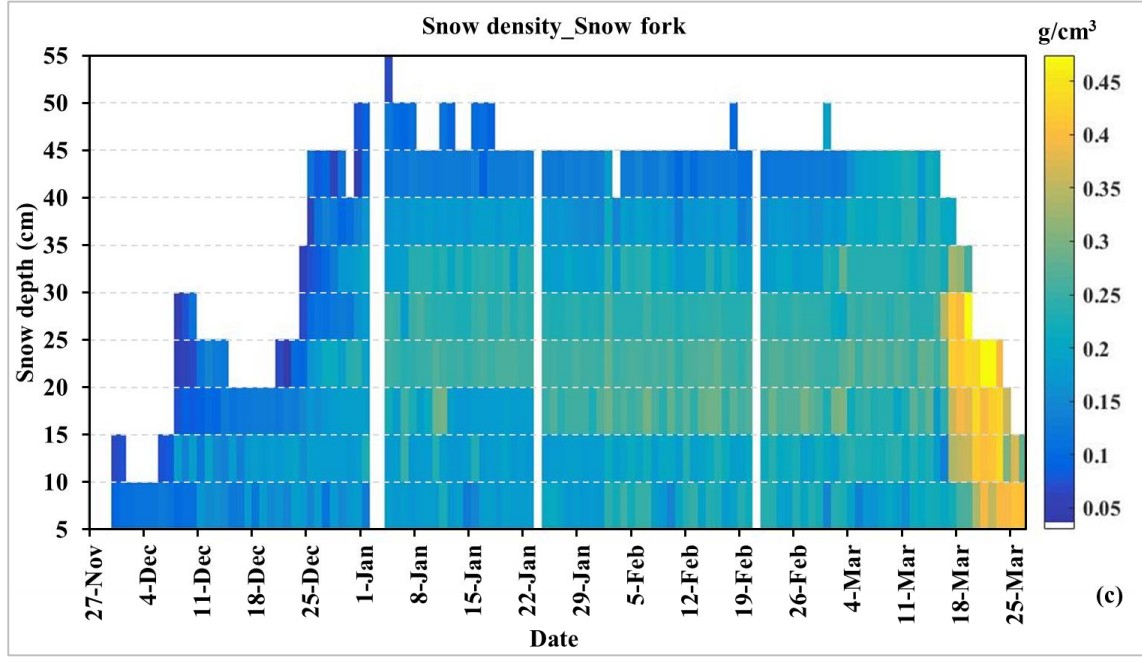

**Figure 9: Daily variation in snow densities measured using three different measurement methods from**
**November 27, 2015 to March 25, 2016. (a) overall snow density measured using snow tube, (b) snow density**
**at 10-cm interval using snow shovel, and (c) snow density at 5-cm interval using snow fork.**

### 4.1.3 Snow temperature

Snow temperature at 0 cm (snow/soil interface temperature) showed little diurnal variation, remaining at approximately -2.0 to 0.7°C. Snow temperature in the top layer had the largest diurnal variation. The diurnal range decreased from top to bottom layers and as the snow depth increased there were more layers with small diurnal variations (Figure 10). After March 17, 2016, the snow temperature of all layers were over 0°C which means snow cover did not refreeze anymore.

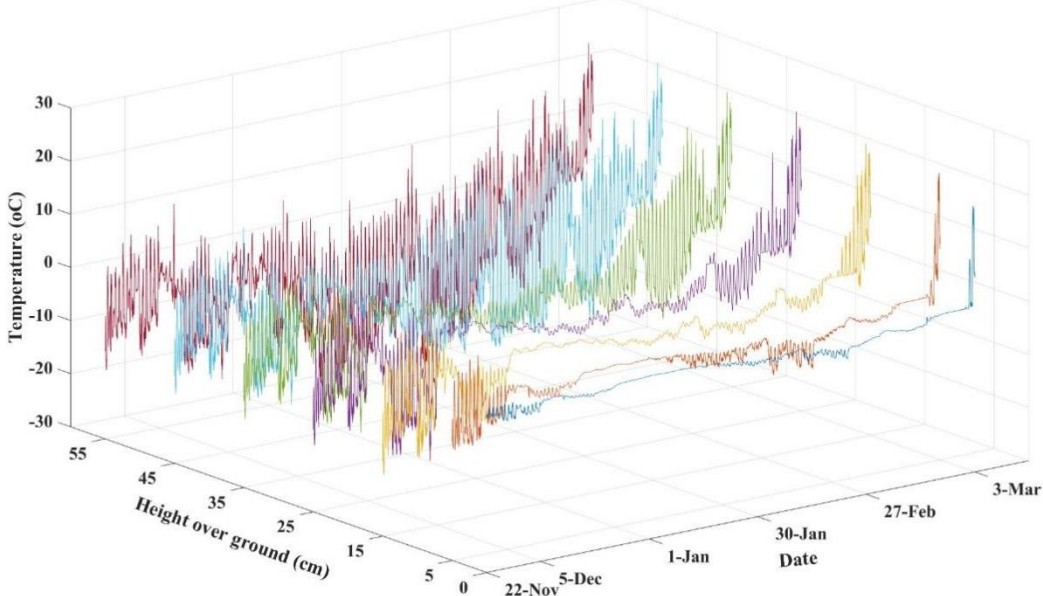

**Figure 10: Minutely variation in layered snow temperatures at 0 cm (snow/soil interface), 5 cm, 15 cm, 25 cm, 35 cm, 45 cm and 55 cm above ground during experiment time.**

### 4.2 Soil temperature and moisture

The soil temperature at 5 and 10 cm remained stable and below 0 °C during the snow season but presented large fluctuation before (after) snow on (off) (Figure 11). The temperature difference between 5 cm and 10 cm was much larger before snow cover onset than during snow cover period. The soil moistures at 10 cm were above 10% before snow cover onset and after snow off, and there were two soil moisture peaks, one from December 12-14 and another from January 1- 20, within the snow cover period.

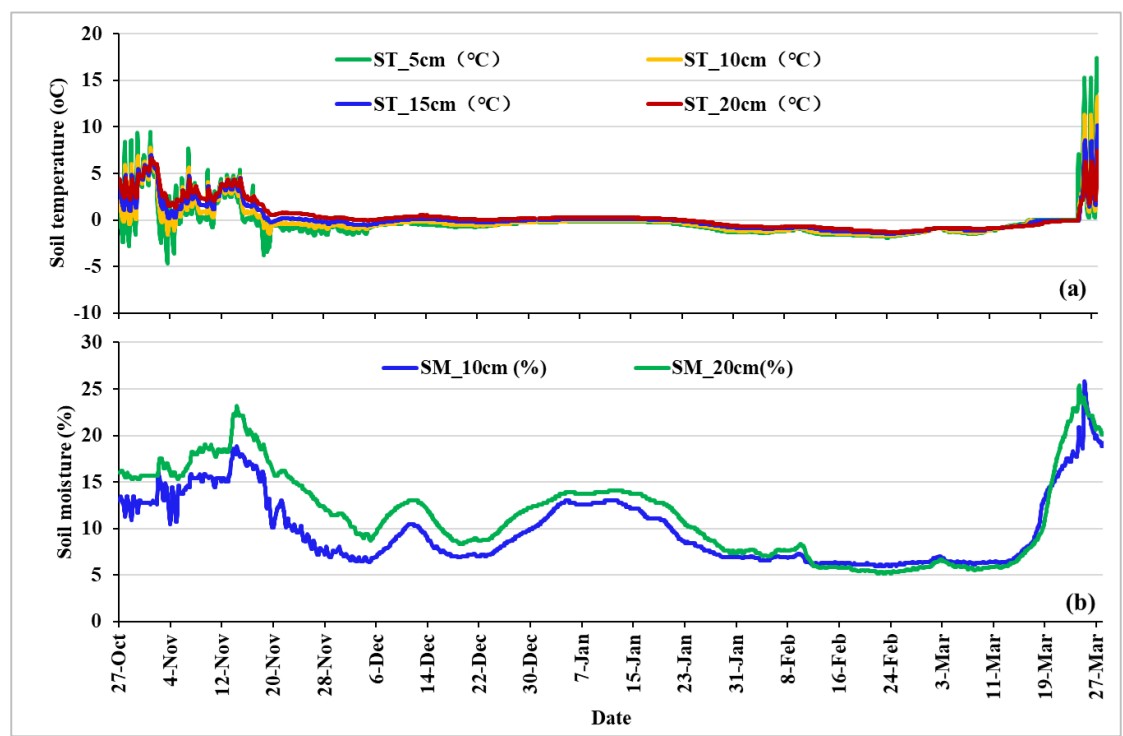


**Figure 11: Hourly soil temperature at 5 cm, 10 cm, 15 cm and 20 cm below the snow/soil interface (a), and soil**
**moisture at 10 cm and 20 cm below the snow/soil interface (b).**
**4.3 Brightness temperature**
The microwave brightness temperatures varied with snow and soil characteristics, and weather
conditions. Figure 12 shows the daily brightness temperatures, brightness temperature difference
between 18 and 36 GHz, and snow depth at 1:00 am local time. Figure 13 shows the hourly variation in
brightness temperatures at 1.4, 18 and 36 GHz and air temperature after February 1. Data show that
Tb36h and Tb36v decreased during the full snow season, Tb18h shows an obvious decline after February
18, and Tb18v after March 3 (Figure 12). After January 4, snow depth stopped increasing, but the
brightness temperature continued to decrease and brightness temperature difference increased. Based on
Figure 8, snow density became stable on January 15. Therefore, after January 4, the decreasing brightness
temperatures was mainly caused by growing grain size.
After February 25, brightness temperature exhibited a distinct cycle of daytime increase and
nighttime decrease (Figure 13), resulting from large liquid water content caused by high daytime air
temperature (above 0$^{\circ}$C) and the melted snowpack refreezing at nighttime. After March 14, there was
another big rise in air temperature and even the nighttime air temperatures were above 0$^{\circ}$C. During this
period of accelerated snowmelt, the liquid water within the snowpack did not refreeze completely at night
and both the brightness temperature and brightness temperature difference exhibited irregular behavior.
The variation of L band was mainly influenced by soil moisture and soil temperature. We have soil
temperatures at 0 cm, 5 cm and 10 cm and soil moisture at 10 cm. However, the L band reflects the soil
moisture within 5 cm which was absent in this experiment. Actually, we did not find the variation of
brightness temperature at L band had relationship with soil moisture at 10 cm and soil temperature.

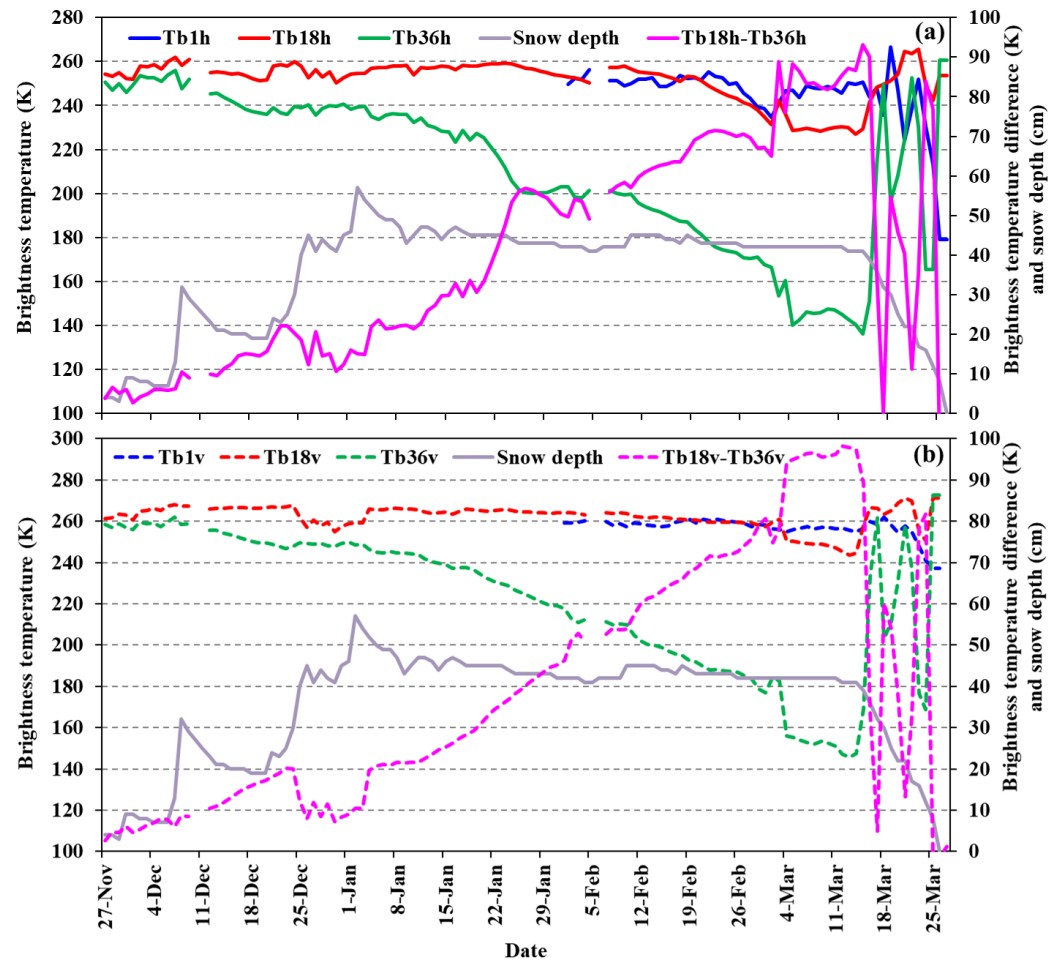


**Figure 12: Daily variations in brightness temperatures at 1.4 GHz, 18 GHz and 36 GHz, for horizontal**
**(Tb1h, Tb18h, Tb36h) and vertical polarizations (Tb1v, Tb18v, Tb36v), and the differences between Tb18h**
**and Tb36h (Tb18h - Tb36h, and between Tb18v and Tb36v (Tb18v - Tb36v), at 1:00 am (local time), from**
**November 27, 2015 to March 26, 2016. (a)for horizontal polarization, and (b) for vertical polarization.**

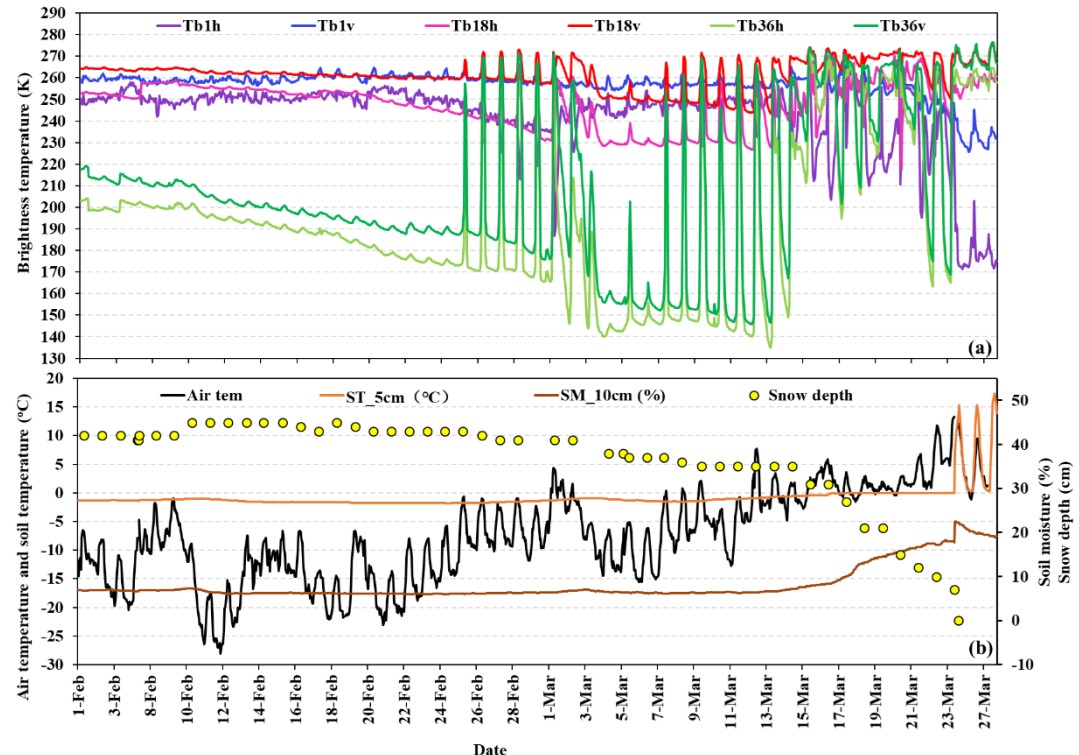


**Figure 13 Hourly variation in Tb1h, Tb18h, Tb36h, Tb1v, Tb18v, and Tb36v (a), air temperature, soil**
**moisture at 10 cm and soil temperature at 5 cm, and daily variation in snow depth (b), from February 1 to**
**March 28, 2016.**

The brightness temperatures at 18.6 and 36.5 GHz from AMSR-2 and at 1.4 GHz from SMAP were
compared with the ground-based observation at the overpass time (Figure 14). Although there were large
differences between satellite and ground-based observations, the general temporal patterns are the same,
even the abrupt change between March 3 and March 4 is captured by both satellite and ground-based
sensors. The correlation coefficients at both polarizations were approximately 0.96, 0.7 and 0.88 for 36
GHz,18.6GHz and 1.4 GHz, respectively. Satellite observed brightness temperature presented less
decrease trend than ground-based observation, and the difference at 36.5 GHz is larger than at 18.6 and
1.4 GHz. Brightness temperatures at 1.4 GHz from both SMAP and ground microwave radiometer kept
stable before March 16, after when, brightness temperature rapidly decreased because of the increase of
liquid water content. The difference between ground-based and satellite observation might be attributed
to the different viewing area.

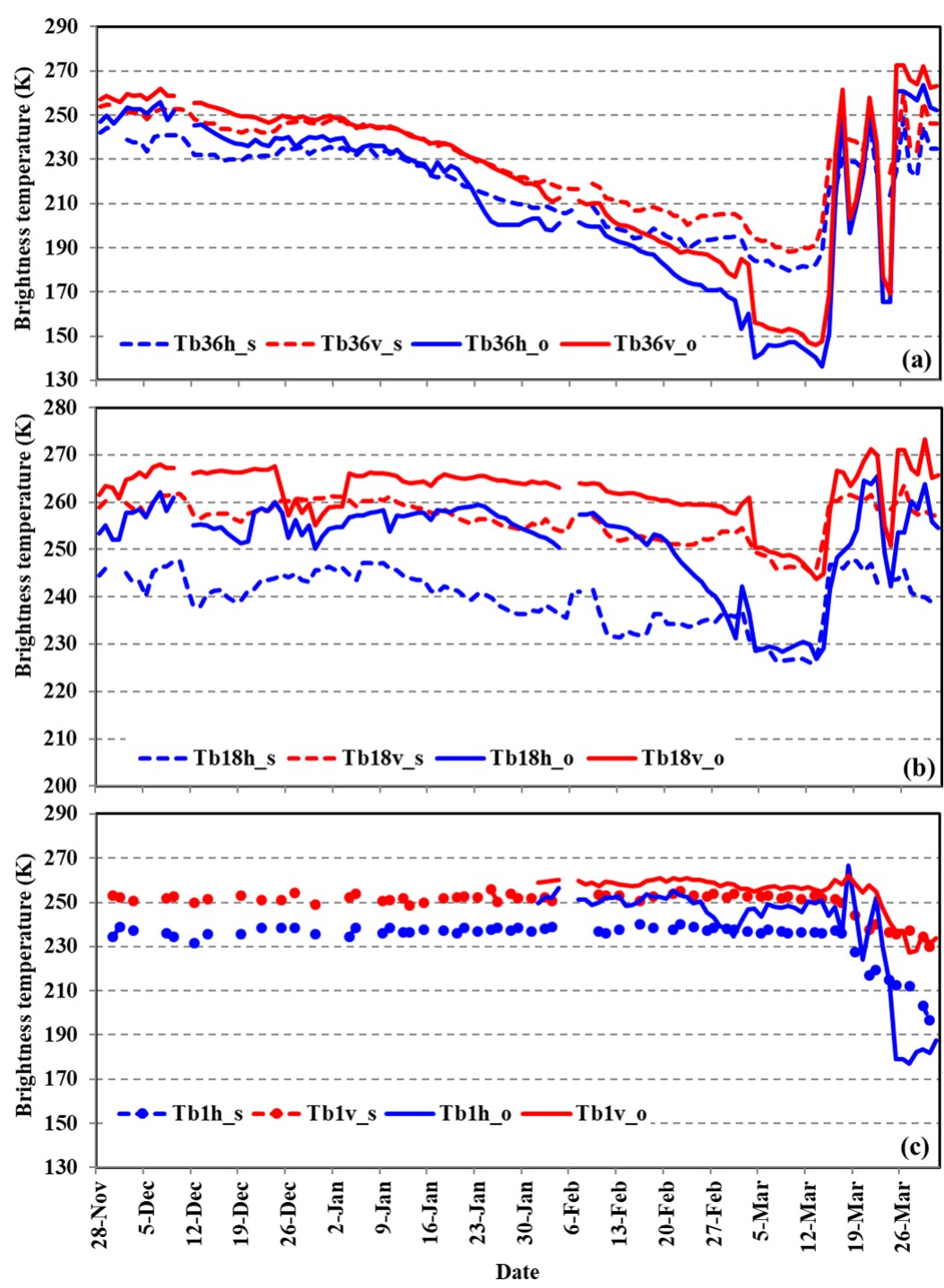

**Figure 14: Comparison of brightness temperature between ground-based and satellite-based observation (s: satellite; o: observation), (a) for 36 GHz, (b) for 18 GHz, (c) for 1.4 GHz**

**4.4 4-component Radiation**

The land surface albedo is strongly related to the land cover. In this experiment, the downward short-wave radiation presented general increase after January, and the trend became distinctive after February (Figure 15). The upward short-wave radiation abruptly increased when the ground was covered by snow (after November 21), and sharply declined on the snow off day (March 25). From the first

snowfall by the end of January, the ratios between upward and downward short-wave radiation were approximately 95%. The ratio decreased with snow age, and in the end of snow season the ratios decreased to below 50% because of increasing melted water.

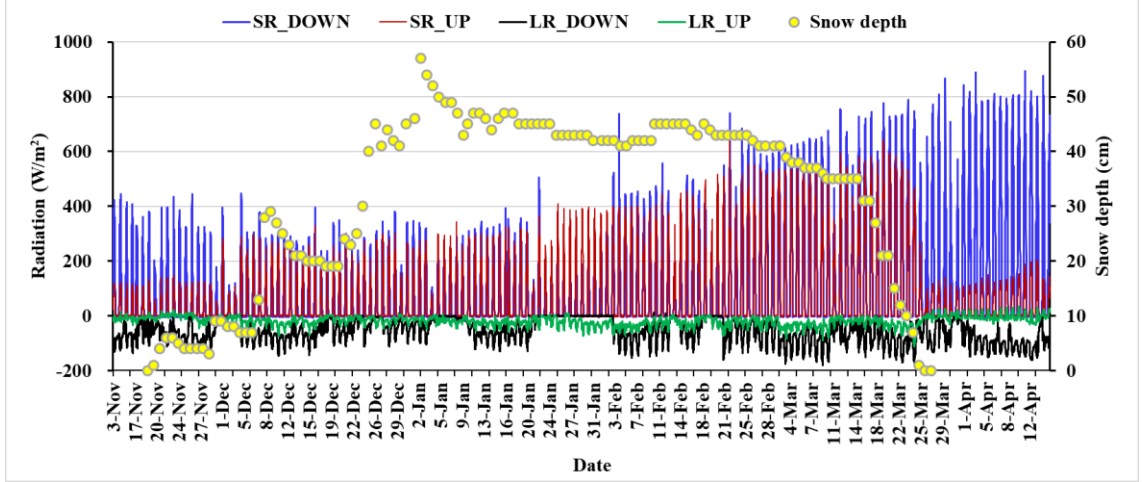

**Figure 15: Minutely variation in 4-component radiation and daily variation in snow depth at Altay station from November 3, 2015 to April 15, 2016.**

**5 Discussion**

5.1 Applications

Although the dataset is just for one season observation, the daily snow pit observation with coincident microwave and optical radiation data in a full snow season provide the most detailed variation of snow parameters which allow researchers to find more details in snow characteristics and their relationship with remote sensing signatures. The dataset also fills the snow observation gap in mid-low snow depth area with relative short snow cover duration.

The snow pit data and microwave brightness temperatures have proven useful for evaluating and updating a microwave emission transfer model of snowpack (Dai et al., 2022). This dataset reflected the general fact that brightness temperature at higher frequencies presented stronger volume scattering of snow grains, and were more sensitive to snow characteristics. This experiment revealed that the dominant control for the variation of brightness temperature was the variation of grain size but not the snow depth. The largest snow depth or SWE did not correspond to the largest brightness temperature difference between 18 and 36 GHz in the condition of dry snowpack. Due to the growth of grain size, the maximum difference occurred before melting for stable snow cover. Therefore, the daily snow depth variations curve derived from passive microwave remote sensing datasets tend to exhibit a temporal offset from those of in situ observation.

During the snow season, brightness temperatures for both polarizations presented similar variations, but they behaved different in some time periods. The horizontal polarization was more sensitive to environment and was less stable than vertical polarization. Besides, the polarization difference at 18 GHz and 36 GHz showed increase and decrease trends, respectively during the experimental period. The results for 18 GHz were opposite to the simulation results (Dai et al., 2022). The different polarization behavior at 18 and 36 GHz might be related to the environmental conditions, snow characteristics and soil conditions. However, the subsurface soil moisture was not observed, the dynamic ground emissivity could not be estimated. L band has strong penetrability, and the brightness temperature variations were

predominantly related to subsurface soil conditions, except when the liquid water content within
snowpack was high. Therefore, in the condition of soil moisture data absence, L band brightness
temperatures were expected to reflect soil moisture variation which influence the soil transmissivity
(Babaeian et al., 2019; Naderpour et al., 2017; Hirahara et al., 2020).
Snow surface albedo significantly influences the incoming solar radiation, playing an important role
in the climate system. The factors altering snow surface albedo contains the snow characteristics (grain
size, SWE, liquid water content, impurities, surface temperature etc), external atmospheric condition and
solar zenith angle (Aoki et al., 2003). Snow albedo was estimated based on snow surface temperatures
in some models (Roesch et al., 1999), while others considered snow surface albedo to depend mainly on
snow aging (Mabuchi et al., 1997). In this experiment, we obtained the 4-component radiation, snow pit
and meteorological data. These data provide nearly all observations of possible influence factors, and
could be utilized to discuss and analyze shortwave radiation process of snowpack, and validate or
improve multiple-snow-layer albedo models.
Snow grain sizes and snow densities within different layers presented different growth rates during
different time periods. Generally, the growth rates are related to the air temperature, pressure and snow
depth (Chen et al., 2020; Essery, 2015; Vionnet et al., 2012; Lehning et al., 2002); therefore, this dataset
can be used to analyze the evolution process of snow characteristics, as well as validation data for snow
models.
5.2 Uncertainties
During the experiment, some uncertainties were produced due to irresistible factors. It is reported
that the sampling depth of the L-band microwave emission under frozen and thawed soil conditions is
determined at 2.5 cm (Zheng et al., 2019). We did not collect subsurface soil moisture, and the L band
radiometer observation began on January 30, 2016. Therefore, it is difficult to obtain the ground
emissivity in the full snow season based on the data. The soil moisture data at 10 and 20 cm under
soil/snow interface cannot be directly used to validate and develop soil moisture retrieval from L band
brightness temperature. We hope detailed soil moisture profile will be observed to estimate the subsurface
soil moisture to fill the gap.
The grain size data were collected through taking photos. When measuring the length of grains, the
grain selection has subjectivity, and the released data are average values. Although the general variation
trend can be reflected by the time series of average grain size, some details might be missed. Therefore,
the original grain photos could be provided through requesting for authors. In snow melt period, large
liquid water content would influence the measurement results of snow fork. So, it is suggested to use
small-size snow shovel or cutter to observe layered snow density in future experiments.
One season observation is quite valuable for developing and validate remote sensing method or
snow model, although the representativeness of this observation remains unknown. We need more years
of observation to endorse or confirm the evolution of snow characteristics.
**6 Conclusions**
In a summary, the IMCS campaign provides a time series of snow pits observation, meteorological
parameters, optical radiation and passive microwave brightness temperatures in the snow season of
2015/2016. The dataset is unique in providing microwave brightness temperatures and coincident daily
snow pits data over a full snow season at a fix site.
The daily snow pit data which provide a detail description of snow grain size, grain shape, snow

density and snow temperature profiles. Generally, grain size grew with snow age, and increased from top to bottom. Snow grains are rounded shape with small grain size in the top layer, and depth hoar with large grain size in the bottom layer. Snow density experienced increase-stable-increase variation, and the densities of the middle layers were greater than the bottom layer due to the well-developed depth hoar in the stable period. The data can be used to analyzes the evolution process of snow characteristics combining with weather data, validate and improve the snow process models, such as SNOWPACK (Lehning et al., 2002), SNTHERM (Chen et al., 2020). The improvement of these models can further enhance the prediction accuracy of land surface process and hydrology models, and the simulation accuracy of snow microwave emission models.

Microwave radiometer data and snow pit data have been utilized to analyze the volume scattering features of snow pack at different frequencies (Dai et al., 2022). Results showed that grain size is the most important factor to influence snow volume scattering. The data can also be used to further analyze polarization characteristics of snow pack combining with soil and weather data, and be used to validate different microwave emission models of snowpack.

The microwave and optical radiations were simultaneously observed. Existing studies reported that the optical equivalent diameter must be used in microwave emission model with caution (Lowe and Picard, 2015; Roy et al., 2013). These data provide a good opportunity to analyze the difference between the influence of grain size on microwave and optical radiation, establishing the bridge between effective optical grain size and microwave grain size.

## 7 Data availability

The IMCS consolidated datasets are available after registration on the National Tibetan Plateau Data Center and available online at http://data.tpdc.ac.cn/zh-hans/data/df1b5edb-daf7-421f-b326-cdb278547eb5/ (doi: 10.11888/Snow.tpdc.270886). Microwave radiometry raw Data are available for scientific use on request from Northwest Institute of Eco-Environment and Resources, Chinese Academy of Sciences.

**Author contributions:** LD and TC designed the experiment. LD, YZ, JT, MA, LX, SZ, YY YH and LX collected the passive microwave and snow pit data. HL provided the 4-component radiation and snow temperature data. LW provided meteorological data. LD write the manuscript, and TC made revision. All authors contributed to the data consolidation.

**Competing interests:** The authors declare that they have no conflict of interest.

**Acknowledgment:** The authors would like to thank the Altay meteorological station for providing logistics service and meteorological data.

**Financial support:** This research was funded by the National Science Fund for Distinguished Young Scholars (grant nos: 42125604), National Natural Science Foundation of China (grant nos: 42171143), and CAS 'Light of West China' Program.

**Appendix**

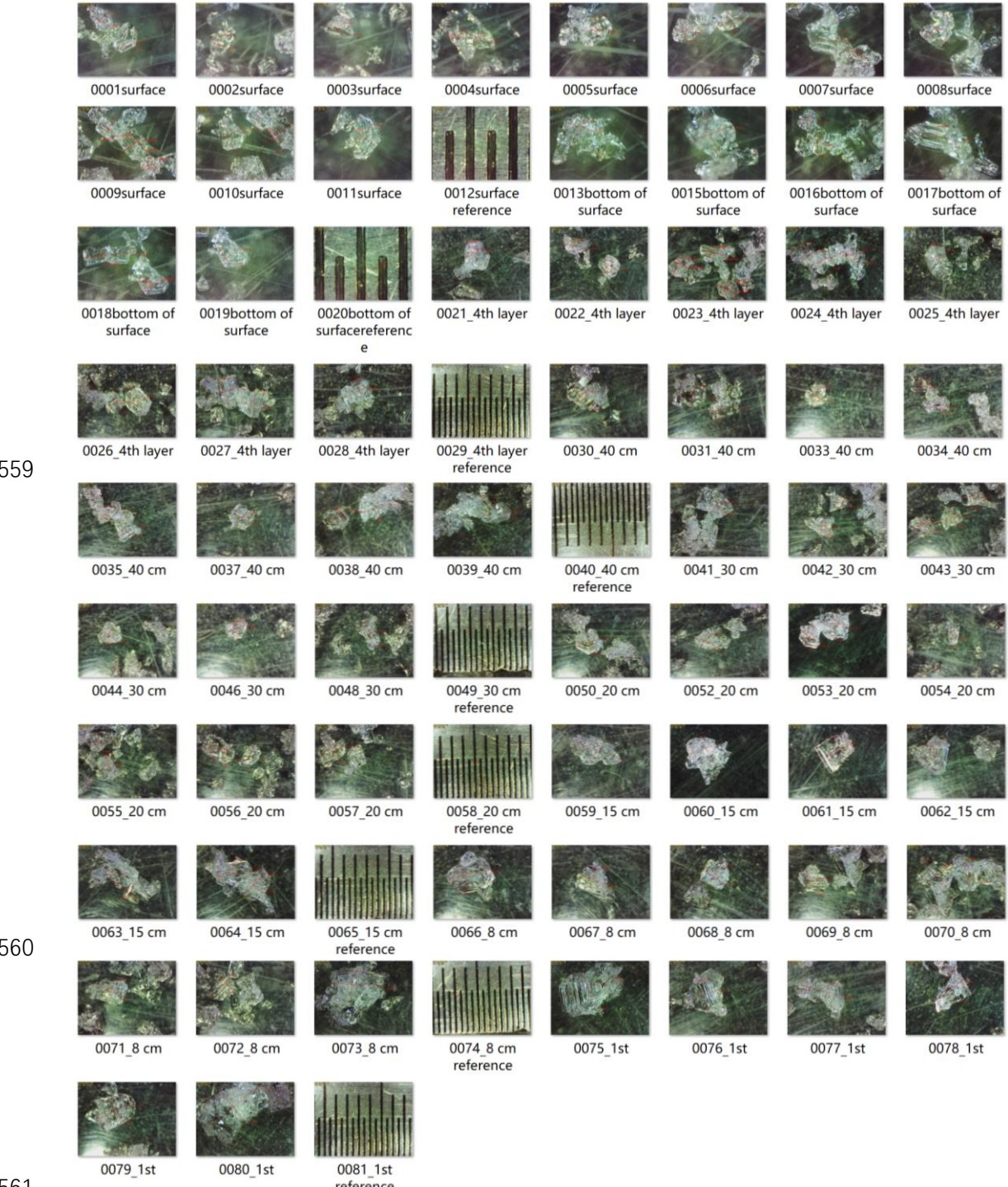

**Figure A1: Photos of grains and reference ruler in each layer on February 15, 2016, and in each photo the longest and shortest axis lengths of the chosen grains are labeled.**

**Table A1. Recorded longest and shortest axis length in Figure A.**

| Stratigraphy | Thickness (cm) | Shape | Grain size (longest axis * shortest axis)(mm) | | | | | | | | |
|---|---|---|---|---|---|---|---|---|---|---|---|
| the fifth | 3cm | #22 | 0.595*0.436 | 0.472*0.471 | 0.450*0.436 | 0.615*0.474 | 0.374*0.314 | 0.647*0.307 | 0.656*0.529 | 0.544*0.519 | 0.717*0.447 |
| | | | 0.750*0.445 | 1.056*0.955 | 0.623*0.378 | 0.451*0.405 | 1.397*0.635 | 1.235*0.327 | 0.600*0.421 | 0.633*0.556 | 0.729*0.423 |
| the fourth | 3cm | #37 | 2.605*2.011 | 1.850*1.328 | 1.626*1.554 | 1.767*1.685 | 1.718*1.535 | 2.255*1.296 | 1.674*1.601 | 1.542*1.269 | 3.505*1.440 |
| | | | 3.055*1.774 | 1.448*1.37 | 2.461*1.914 | 2.757*2.115 | 2.179*2.059 | 2.393*1.788 | | | |
| | | | | | | | | | | | |
| the third | 25cm | #27, #31, #37 | 2.569*1.607 | 2.073*2.130 | 2.591*1.414 | 1.869*1.802 | 2.067*1.266 | 1.209*1.106 | 1.719*1.188 | 1.648*0.975 | 1.911*1.582 |
| | | | 1.921*1.710 | 1.518*1.067 | 1.291*1.147 | 1.690*1.551 | 1.756*1.398 | 1.812*1.263 | 1.733*1.672 | 1.880*1.518 | 2.411*1.220 |
| | | | 2.118*1.727 | 1.614*1.457 | 1.795*1.705 | 2.215*2.311 | 1.864*1.692 | 1.967*1.651 | 2.008*1.395 | 1.362*1.141 | 1.484*1.291 |
| the second | 12 | #33, #34 | 4.251*2.266 | 3.012*2.65 | 2.805*1.995 | 1.799*1.415 | 1.402*1.195 | 3.040*2.073 | 2.850*2.095 | | |
| | | | 3.900*2.532 | 2.420*2.333 | 2.515*2.206 | 2.044*2.032 | 2.506*2.363 | 2.894*2.161 | 2.413*1.950 | 2.494*1.816 | 4.929*3.257 |
| the first | 4 | #40, #34, #38 | 4.933*3.378 | 3.207*2.774 | 3.562*1.701 | 2.818*1.668 | 3.581*2.518 | 6.179*3.562 | | | |
| | | | | | | | | | | | |

**Table A2: One example of record table for snow density observation.**

| observation date: | 20160111 | | observation time: 9:03-9:40 | | weather: clear | | snow depth: 48cm |
|---|---|---|---|---|---|---|---|
| **Snow Folk table** | | | **Snow tube table** | | | | |
| observation height(cm) | liquid water content(%) | snow density(g/cm3) | snow depth(cm) | 46.5 | | 47 | 47.5 |
| | 0 | 0.1923 | snow pressure(g/cm2) | 9.1 | | 9 | 9.5 |
| 5 | 0.118 | 0.1882 | snow density(g/cm3) | 0.1957 | | 0.1915 | 0.2000 |
| | 0 | 0.1882 | | | | | |
| | 0.461 | 0.164 | **snow shovel table** | | | | |
| 10 | 0.46 | 0.1631 | observation layer (cm) | weight of shovel+snow(g) | weight of shovel(g) | | snow density(g/cm3) |
| | 0.461 | 0.1361 | | 865.04 | 572.16 | | 0.1953 |
| | 0.123 | 0.2532 | 0-10 | 858.72 | 572.16 | | 0.1910 |
| 15 | 0 | 0.2506 | | 866.69 | 572.16 | | 0.1964 |
| | 0 | 0.2417 | | 878.58 | 572.16 | | 0.2043 |
| | 0.24 | 0.2159 | 10-20 | 887.04 | 572.16 | | 0.2099 |
| 20 | 0.119 | 0.2155 | | 872.79 | 572.16 | | 0.2004 |
| | 0.119 | 0.2146 | | 905.34 | 572.16 | | 0.2221 |
| | 0.117 | 0.1977 | 20-30 | 903.41 | 572.16 | | 0.2208 |
| 25 | 0 | 0.1994 | | 907.88 | 572.16 | | 0.2238 |
| | 0 | 0.1984 | | 832.75 | 572.16 | | 0.1737 |
| | 0 | 0.1919 | 30-40 | 838.14 | 572.16 | | 0.1773 |
| 30 | 0 | 0.1966 | | 837.27 | 572.16 | | 0.1767 |
| | 0 | 0.1928 | | | | | |
| | 0 | 0.1534 | 40-50 | | | | |
| 35 | 0 | 0.1517 | | | | | |
| | 0 | 0.1472 | | | | | |
| | 0.325 | 0.1097 | 50-60 | | | | |
| 40 | 0 | 0.1054 | | | | | |
| | 0.107 | 0.1088 | | | | | |
| | 0 | 0.0922 | | | | | |
| 45 | 0 | 0.0991 | | | | | |
| | 0 | 0.0928 | | | | | |
| 50 | | | | | | | |

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
