# Peer review of "Microwave radiometry experiment for snow in Altay"

_Earth System Science Data, 2022_

## Author Comment (AC1)

This manuscript introduces a dataset collected from an Integrated Microwave Radiometry Campaign for snow (IMCS) conducted at the Altay National Reference Meteorological station (ANRMS) in Xinjiang, China. The dataset could be very useful for the evaluation and development of microwave and optical radiative transfer models and snow evolution process models.

The topic of the study is interesting and well fits the scope of the journal, especially for this special issue. The manuscript is well written, logically organized, and the details of field campaign are easy to follow. The data processing is careful and well documented. However, there are still some concerns that need to be addressed. Thus, I am supportive of the publication after a minor revision to further improve the quality or make it more clear for the readers to understand the results. Below are my suggestions:

Thanks for these constructive suggestions. We have revised the manuscript according to your comments.

**General comments**

1. Title: change "in situ time series of data" to "time series of in situ data"; delete "and environment".

   Re: Thanks for the recommendation. We revised it.

2. L59-L89: It's suggested to provide a table to summarize the main characteristics of those mentioned experiments and the experiment presented in this paper.

   Re: Thanks for the suggestion.

   We added a table to summarize the five experiments.

   These experiments are summarized in table 1.

   Table 1 Summary of existing experiments for microwave and optical radiation and physical feasutes of snowpack

| Campaign | Location | Temporal range | Observation content |
| --- | --- | --- | --- |
| CLPX | Different sites in Colorado, | February and March of 2002 and 2003 | Inconsecutive multiple sensor observation, including microwave radiometry over snow, and matched snow pit measurements were conducted at different sites with short temporal range. |
| SnowEx-year 1 | Grand Mesa, and Senator Beck Basin, Colorado | February of 2017 | Inconsecutive multiple sensor observation, including microwave radiometry over snow, and matched snow pit measurements were conducted at different sites with short temporal range. |

| CMRES[1] | Mobile observation at Forest, open and lake in the northern Canadian region | November of 2009-April of 2010 | Mobile microwave radiometry and snow pit observation within footprint of radiometer. Short temporal range and inconsecutive observation |
|---|---|---|---|
| NoSREx | Fixed site in Sodankylä, Finland | Snow season during 2009-2013 | Consecutive microwave radiometry and SAR observation over snow, and weekly snow pit measurement |
| JERBS[2] | Fixed site in Japan | Snow season during 1999-2000 | Consecutive optical radiation observation over snow and consecutive snow pit measurement at 3 or 4-day interval. |
| IMCS | Fixed site in China | November of 2015-March of 2016 | Consecutive microwave radiometry and optical radiation observation, and consecutive daily snow pit measurements. |

Note: [1]CMRES: Microwave radiometry experiment on snow cover conducted in northern Canada

[2]JERBS: Experiment of radiation budget over snow cover in Japan

L115-116: Did the author measure the surface heat flux, e.g. sensible and latent heat flux?

Re: No, we did not set up instrument to measure the surface heat flux, and the data collected by the Altay meteorological station also did not cover it.

3. Figure 1: the pictures in the blue, red and pink boxes are too small to identify the exact instrument. Maybe the authors can divide this figure to two figures.

Re: The purpose of Figure 1 is to describe the measurement position of all parameters. The instruments used for measuring parameters in this study in blue and red boxes are clearly showed in figure 4 and figure 5. The microwave radiometer in pink box was magnified as a picture on the upper left corner. In order to make the microwave radiometer more clear, we enlarged the content of red box and insert the enlarged picture in section 2.3.1.

[Figure]

Figure 2 Ground-based microwave radiometer observation.

4. L151: It's suggested to merge Section 2.2 and 2.3, and the presentation can be grouped by the measurement parameters, e.g. microwave radiometry, snow pit...

Re: Thanks for this suggestion, we merged section 2.2 and 2.3 to section 2.2 Measurement methods, and instruments were separately described in subsections of matched parameters.

Please see detailed revision in section 2.2 in the revised manuscript.

5. L183-198: what's the calibration accuracy for the microwave radiometry? Incidence angle of the radiometry measurement should be provided. It seems too large for the sky temperatures at L-band which is generally around ~5K.

Re:

1. In order to fulfil the requirement of low maintenance regarding absolute calibrations, the instrument is equipped with a two-stage thermal control system for all receivers with an accuracy of ±0.05 K over the full operating temperature range. The calibration accuracy for the microwave radiometry is 1K. The differences between before-calibration and after-calibration Tb values were within 1K for L band, 0.5 K for K and Ka bands.

The sky calibrations were performed under the clear sky condition. During the experiment, we did multiple times of sky calibration. The L band radiometer didn't work at the beginning of the experiment. We contacted Germany company to solve the problem. It took a long time to fix it, and the tb at 1.4 GHz were obtained from 30, January, 2016. So, two sky calibration was for L band, and they were performed at 3, February and 6, March. However, the values changed largely. On 3, February, sky Tb at L band were 7-8K, and 15-16K, for horizontal and vertical polarization, respectively. on 6, March, they are -1~3K and 1~5 K.

However, on March 27 and 31 when there is no snow cover, we did another sky scanning, the brightness temperatures at L band were -1~1 and 5~8K for horizontal and vertical polarization, respectively. We also doubted it, but the objective of this experiment focus on snow cover, L band showed little sensitive to snow characteristics, so, we did not deeply consider this problem.

We revised this sentence to describe the problem, so data user will consider it.

"This radiometer was sky tipping calibrated. In the clear sky conditions, the sky brightness temperatures are approximately 7.8$\pm$1K and 15.7$\pm$0.7K at 1.4 GHz for horizontal polarization and vertical polarization, respectively; those were approximately 29.7$\pm$0.3 K and 29.3$\pm$0.9 K at 18.7GHz and 36.5 GHz, respectively."

Was revised to

L176-182: " The microwave radiometers at K and Ka bands began working from November 27, 2015, but the L band radiometer did not work until January 30, 2016.These radiometers were sky tipping calibrated, and the calibration accuracy is 1 K. In clear sky conditions, the sky brightness temperatures were approximately 29.7$\pm$0.3 K at 18.7 GHz for both polarizations and 29.3$\pm$0.9 K at 36.5 GHz for both polarizations. But the sky brightness temperature at L band showed large fluctuation. They ranged from -1 to 8 K for horizontal polarization, and 1 to 16 K for vertical polarization."

2. in this study, fixed incidence observations were conducted every day, and the fixed incidence angle is 50$^o$. multi-angle observations were conducted on 17 days, and angles include 30, 35,40, 45, 50, 60$^o$.

6. Figures 6/8/9: These figures can be improved, it's difficult to distinguish the lines.

Re: they are revised to make them more clear.

Figure 6:

[Figure]

Was revised to

[Figure]

Figure 7: Daily variation in snow layers and grain shape in each layer from November 27, 2015 to March 25, 2016.

Figure 8:

There are 10 layers of snow density in figure 8(bf), so the lines are difficult to distinguish. The folding line figure was changed to image figure.

[Figure]

Was revised to

[Figure]

The line colors in Figure 8(b) were changed to make them more distinguishable.

[Figure]

Was revised to

[Figure]

Figure 9: we changed the style of figure 9, and considering another reviewer's suggestion, we simplified the timestamp for x-axis.

[Figure]

Was revised to

[Figure]

Figure 10: Minutely variation in layered snow temperatures at 0 cm (snow/soil interface), 5 cm, 15 cm, 25 cm, 35 cm, 45 cm and 55 cm above ground during experiment time.

7. Figure 10: This figure can be divided into two figures for the soil moisture and temperature, respectively.

Re: We divided the figure into two figures, and considering another reviewer's suggestion, we simplified the timestamp for x-axies.

[Figure]

Was revised to

[Figure]

Figure 11: Hourly soil temperature at 5 cm,10 cm, 15 cm and 20 cm below the snow/soil interface (a), and soil moisture at 10 cm and 20 cm below the snow/soil interface (b).

8. Figure 11: I suggest this figure can be divided into two figures. Specifically, Figure 11a can be divided into two figures for the H- and V- polarizations, respectivley. Figure 11b can be another figure, and the whole study period can be divided into several periods for the H- and V- polarizations, respectivley. For example, it can be freezing, thawing periods, and it's suggested to include the snow, soil moisture and temperautre measurements to show the link between these measurements with the diurnal variations of brightness temperature. Besides, what can be the reason cause the large variaitons found around 2016/2/25 and 2016/3/23?

Re:

1. Figure 11a was divided into two figure. One for H polarization, another for V polarization.

[Figure]

Figure 12: Daily variations in brightness temperatures at 1.4 GHz, 18 GHz and 36 GHz, for horizontal (Tb1h, Tb18h, Tb36h) and vertical polarizations (Tb1v, Tb18v, Tb36v), and the differences between Tb18h and Tb36h (Tb18h - Tb36h, and between Tb18v and Tb36v (Tb18v - Tb36v), at 1:00 am (local time), from November 27, 2015 to March 26, 2016. (a)for horizontal polarization, and (b) for vertical polarization.

2. Figure 11a shows the brightness temperature through the whole snow season. Figure 11b focus on the melting phase. According to the comments, we added the variation in snow depth, soil moisture and soil temperature to link the variation in different parameters.

Figure 11b

[Figure]

was revised to:

[Figure]

Figure 13 Hourly variation in Tb1h, Tb18h, Tb36h, Tb1v, Tb18v, and Tb36v (a), air temperature, soil moisture at 10 cm and soil temperature at 5 cm, and daily variation in snow depth (b), from February 1 to March 28, 2016.

3.    Large variation around 2016/2/25 and 2016/3/23? From the figure, there is no large variation around 2016/2/25, but around 2016/3/1. The reason maybe the continuous melting-refreezing resulting in abrupt increase of grain size. On

2016/3/1 and 2016/3/2, the maximum air temperature increased over 273K, and large melting occurred. The air temperature decreased in the following several days resulted in large increase in grain size. After 2016/3/15, the melting snow would not refreeze at nighttime, so the brightness temperature cannot reflect the scattering of snow grains, and was controlled by liquid water; thus, presenting desultorily fluctuation.

9. Figure 12: It's also suggested to compare the in situ measurements with the SMAP satellite measurements for the 1.4 GHz.

   Re: Thanks for the suggestion. We added the comparison of 1.4 GHz. 1.4 GHz presents similar variation trend, especially in the snow melt period.

[Figure]

10. Figure 13: it's difficult to distinguish the lines. Maybe you can put the shortwave radiation in one figure (e.g. 13a), and the longwave radiation in the other figure (e.g. 13b). Also, it's suggested to include the snow measurements to show the impact of snow on these measurements.

    Re: it will be more convenient for comparison to put the 4-component in a figure. In order to make them more clear, we changed the line color, and added daily snow depth in the figure.

[Figure]

was revised to

[Figure]

Figure 15: Minutely variation in 4-component radiation and daily variation in snow depth at Altay station from November 3 2015 to April 15 2016.

11. Figure A1: the figure is too small, maybe you can increase the row to cover the full page. Also, some characters are difficult to understand (it seems to be Chinese).

Re: Thanks, we translate the Chinese words, and the photos were

[Figure]

Was revised to

12. Table A2: the figure is too small.

Re:

**observation date:** 20160111  **observation** 19:03-9:40  **weather:** clear  **snow depth:** 48cm

**Snow Folk table**

| observation height (cm) | liquid water content (%) | snow density (g/cm3) |
|---|---|---|
| 5 | 0 | 0.1923 |
|  | 0.118 | 0.1882 |
|  | 0 | 0.1882 |
| 10 | 0.461 | 0.164 |
|  | 0.46 | 0.1631 |
|  | 0.461 | 0.1361 |
| 15 | 0.123 | 0.2532 |
|  | 0 | 0.2506 |
|  | 0 | 0.2417 |
| 20 | 0.24 | 0.2159 |
|  | 0.119 | 0.2155 |
|  | 0.119 | 0.2146 |
| 25 | 0.117 | 0.1977 |
|  | 0 | 0.1994 |
|  | 0 | 0.1984 |
| 30 | 0 | 0.1919 |
|  | 0 | 0.1966 |
|  | 0 | 0.1928 |
| 35 | 0 | 0.1534 |
|  | 0 | 0.1517 |
|  | 0 | 0.1472 |
| 40 | 0.325 | 0.1097 |
|  | 0 | 0.1054 |
|  | 0.107 | 0.1088 |
| 45 | 0 | 0.0922 |
|  | 0 | 0.0991 |
|  | 0 | 0.0928 |
| 50 |  |  |
| 55 |  |  |

**snow tube table**

| snow depth(cm) | 46.5 | 47 | 47.5 |
|---|---|---|---|
| snow pressure(g/cm2) | 9.1 | 9 | 9.5 |
| snow density(g/cm3) | 0.1957 | 0.1915 | 0.2000 |

**snow shovel table**

| observation layer (cm) | weight of shovel+snow(g) | weight of shovel(g) | snow density(g/cm3) |
|---|---|---|---|
| 0-10 | 865.04 | 572.16 | 0.1953 |
|  | 858.72 | 572.16 | 0.1910 |
|  | 866.69 | 572.16 | 0.1964 |
| 10-20 | 878.58 | 572.16 | 0.2043 |
|  | 887.04 | 572.16 | 0.2099 |
|  | 872.79 | 572.16 | 0.2004 |
| 20-30 | 905.34 | 572.16 | 0.2221 |
|  | 903.41 | 572.16 | 0.2208 |
|  | 907.88 | 572.16 | 0.2238 |
| 30-40 | 832.75 | 572.16 | 0.1737 |
|  | 838.14 | 572.16 | 0.1773 |
|  | 837.27 | 572.16 | 0.1767 |
| 40-50 |  |  |  |
| 50-60 |  |  |  |

Was revised to

**observation date:** 20160111  **observation time:** 9:03-9:40  **weather: clear**  **snow depth: 48cm**

**Snow Folk table**

| observation height (cm) | liquid water content(%) | snow density (g/cm3) |
|---|---|---|
| 5 | 0 | 0.1923 |
|  | 0.118 | 0.1882 |
|  | 0 | 0.1882 |
| 10 | 0.461 | 0.164 |
|  | 0.46 | 0.1631 |
|  | 0.461 | 0.1361 |
| 15 | 0.123 | 0.2532 |
|  | 0 | 0.2506 |
|  | 0 | 0.2417 |
| 20 | 0.24 | 0.2159 |
|  | 0.119 | 0.2155 |
|  | 0.119 | 0.2146 |
| 25 | 0.117 | 0.1977 |
|  | 0 | 0.1994 |
|  | 0 | 0.1984 |
| 30 | 0 | 0.1919 |
|  | 0 | 0.1966 |
|  | 0 | 0.1928 |
| 35 | 0 | 0.1534 |
|  | 0 | 0.1517 |
|  | 0 | 0.1472 |
| 40 | 0.325 | 0.1097 |
|  | 0 | 0.1054 |
|  | 0.107 | 0.1088 |
| 45 | 0 | 0.0922 |
|  | 0 | 0.0991 |
|  | 0 | 0.0928 |
| 50 |  |  |

**Snow tube table**

| snow depth(cm) | 46.5 | 47 | 47.5 |
|---|---|---|---|
| snow pressure(g/cm2) | 9.1 | 9 | 9.5 |
| snow density(g/cm3) | 0.1957 | 0.1915 | 0.2000 |

**snow shovel table**

| observation layer (cm) | weight of shovel+snow(g) | weight of shovel(g) | snow density(g/cm3) |
|---|---|---|---|
| 0-10 | 865.04 | 572.16 | 0.1953 |
|  | 858.72 | 572.16 | 0.1910 |
|  | 866.69 | 572.16 | 0.1964 |
| 10-20 | 878.58 | 572.16 | 0.2043 |
|  | 887.04 | 572.16 | 0.2099 |
|  | 872.79 | 572.16 | 0.2004 |
| 20-30 | 905.34 | 572.16 | 0.2221 |
|  | 903.41 | 572.16 | 0.2208 |
|  | 832.75 | 572.16 | 0.1737 |
| 30-40 | 838.14 | 572.16 | 0.1773 |
|  | 837.27 | 572.16 | 0.1767 |
| 40-50 |  |  |  |
| 50-60 |  |  |  |

13. There were other microwave radiometry experiments conducted in the Third Pole, and the authors are suggested to include it in the Introduction part. Please find below several references for the details.

Zheng, D., Li, X., Wen, J., Hofste, J.G., van der velde, R., Wang, X., Wang, Z., Bai, X., Schwank, M., and Su, Z. (2022). Active and Passive Microwave Signatures of Diurnal Soil Freeze-Thaw Transitions on the Tibetan Plateau. IEEE

Transactions on Geoscience and Remote Sensing, 60, doi: 10.1109/TGRS.2021.3092411.

Zheng, D., Li, X., Zhao, T., Wen, J., van der Velde, R., Schwank, M., Wang, X., Wang, Z., and Su, Z. (2021). Impact of Soil Permittivity and Temperature Profile on L-Band Microwave Emission of Frozen Soil. IEEE Transactions on Geoscience and Remote Sensing, 59(5), 4080-4093.

Zhang, P., Zheng, D.*, van der Velde, R., Wen, J., Zeng, Y., Wang, X., Wang, Z., Chen, J., and Su, Z.* (2021). Status of the Tibetan Plateau observatory (Tibet-Obs) and a 10-year (2009–2019) surface soil moisture dataset. Earth Syst. Sci. Data, 13, 3075–3102.

Zheng, D., Li, X., Wang, X., Wang, Z., Wen, J., van der Velde, R., Schwank, M., and Su, Z. (2019). Sampling depth of L-band radiometer measurements of soil moisture and freeze-thaw dynamics on the Tibetan Plateau. Remote Sensing of Environment, 226, 16-25.

Re: Thanks for the remind. We added introduction of these microwave radiometry experiments.

A paragraph was added in section introduction, and references were added in section reference.

L89-93: In the Tibetan plateau with shallow snow cover, multiple years of microwave radiometry observation at L band were conducted to study passive microwave remote sensing of frozen soil (Zheng et al., 2019, 2021a and 2021b). However, in the long term series of experiment, no snow pit was measured and the microwave radiometry observation was performed at L band which is insensitive to snowpack.

Zheng, D., Li, X., Zhao, T., Wen, J., van der Velde, R., Schwank, M., Wang, X., Wang, Z., and Su, Z. : Impact of Soil Permittivity and Temperature Profile on L-Band Microwave Emission of Frozen Soil. IEEE Transactions on Geoscience and Remote Sensing, 59(5), 4080-4093, DOI: 10.1109/TGRS.2020.3024971, 2021.

Zhang, P., Zheng, D., van der Velde, R., Wen, J., Zeng, Y., Wang, X., Wang, Z., Chen, J., and Su, Z.: Status of the Tibetan Plateau observatory (Tibet-Obs) and a 10-year (2009–2019) surface soil moisture dataset. Earth Syst. Sci. Data, 13, 3075–3102, https://doi.org/10.5194/essd-13-3075-2021, 2021.

Zheng, D., Li, X., Wang, X., Wang, Z., Wen, J., van der Velde, R., Schwank, M., and Su, Z.: Sampling depth of L-band radiometer measurements of soil moisture and freeze-thaw dynamics on the Tibetan Plateau. Remote Sensing of Environment, 226, 16-25, doi.org/10.1016/j.rse.2019.03.029, 2019.

15. Grammar check:

L33: change "sow" to "snow"

Re: it was corrected.

L40: delete "and optical"; delete "evolution"

 Re: it was revised.

**Comments on the Dataset**

L41: the link to the dataset cannot be open, please provide the detailed download link.

Re: Sorry for failing to open the link.

The link was revised to

L41 and L538: http://data.tpdc.ac.cn/zh-hans/data/df1b5edb-daf7-421f-b326-cdb278547eb5/ (doi: 10.11888/Snow.tpdc.270886.)

---

## Author Comment (AC2)

**Microwave radiometry experiment for snow in Altay China: in situ time series of data for electromagnetic and physical features of snow pack and environment**

**Dai et al. 2022**

This manuscript presents a comprehensive dataset of snowpack physical characteristics from a single site for one snow season. The dataset includes microwave and optical radiation data, traditional physical characteristics measured from snow pits, meteorological observations and soil conditions. The dataset contains the variables required for most physically-based snowpack models. In general, the authors do a nice job of describing what was done and why. The manuscript is well prepared and easy to follow, however, it would benefit from English language editing. There is also a fair amount of repetition and the article could be condensed for improved readability.

While I do not have any major concerns, I have a few minor comments. Additionally, I have provided a number of minor editorial suggestions for the authors to consider.

Re: Thank you very much for your comments and constructive recommendations. Your detail revisions and correction on sentence organization and language grammar largely improved readability of this paper. Another reviewer also pointed out the repetition problem, and suggest merging section 2.2 and 2.3. We reorganized section 2.2. and 2.3 in this revised manuscript.

According to your recommendation, the dataset was also reorganized into NetCDF format which presented data more clear.

1. **Data access:** I was unable to access the data directly using the links in the manuscript. I was able to access the data here https://data.tpdc.ac.cn/en/data/df1b5edb-daf7-421f-b326-cdb278547eb5/ , using the doi as a search term.

   Re: Sorry for failing to open the link.

   The link was revised to

   L41 and L538: http://data.tpdc.ac.cn/zh-hans/data/df1b5edb-daf7-b326-cdb278547eb5/ (doi: 10.11888/Snow.tpdc.270886.)

2. The authors describe the dataset as a 'consolidated' dataset. I am not sure 'consolidated' is the best term to describe it. The dataset is comprised of numerous asci files and excel spreadsheets in various directories. It is more of an 'assembled' dataset. There was some 'consolidation' when multiple observations were averaged but to me that is part of the natural data management process.

Re: Yes, we agree with you. We adopted the third comments to consolidate daily data into a single file. The minutely, ten-minute, and hour data were also consolidated into NetCDF files. The data released at the national Tibetan plateau data center, China were updated according to the new files.

3. Did the authors consider any other file formats such as NetCDF or data management strategies? For example, could the daily measurements not have been consolidated into a single netcdf file? I found the various directories and files a bit cumbersome.

Re: Thanks for this constructive suggestion. Daily data, ten-minute data, hourly data, minutely data were all separately consolidated into NetCDF files. The data descriptions were also updated. Please see section 3 and table 3.

**Table 3 Description of consolidated data**

| Data | Content | File name | Variables |
|------|---------|-----------|-----------|
| Brightness temprature | Brightness temperature | TBdata.nc | Year, month, day, hour, minute, second, Tb1h, Tb1v, Tb18h, Tb18v, Tb36h, Tb36v, incidence angle, azimuth angle |
| | Multi-angle brightness temperatures | TBdata-multiangle.nc | Year, month, day, hour, minute, second, Tb1h, Tb1v, Tb18h, Tb18v, Tb36h, Tb36v, incidence angle, azimuth angle |
| Manual snow pit data | Layer thickness, layered grain size and shape, snow density | Daily snow pit data.nc | Year, month, day, snow depth, th1, Lg1, Sg1, th2, Lg2, Sg2, th3, Lg3, Sg3, th4, Lg4, Sg4, th5, Lg5, Sg5, th6, Lg6, Sg6, Stube, SS_0-10, SS_10-20, SS_20-30, SS_30-40, SS_40-50, SF_5, SF_10, SF_15, SF_20, SF_25, SF_30, SF_35, SF_40, SF_45, SF_50, shape1, shape2, shape3, shape4, shape5, shape5 |
| Automated snow temperature and radiation data | 4-component radiation, snow temperature | Ten-minute 4 component radiation and snow temperature.nc | Year, month, day, hour, minute, SR_DOWN, SR_UP, LR_DOWN, LR_UP, T_Sensor, ST_0cm, ST_5cm, ST_15cm, ST_25cm, ST_35cm, ST_45cm, ST_55cm |
| Meteorological and soil data | meteorological data, soil moisture and temperature | Hourly meteorological and soil data.nc | Year, month, day, hour, Tair, Wair, Pair, Win, SM_10cm, SM_20cm, Tsoil_5cm, Tsoil_10cm, Tsoil_15 cm, Tsoil_20cm |

Note: th: snow thickness, Lg: long axis, Sg: short axis, shape: grain shape;
Stube: snow density observed using snow tube, SS: snow density observed using snow shovel, SF: snow density observed using snow fork; ST: snow temperature; SR_DOWN: downward short-wave radiation, SR_UP: upward short-wave radiation, LR_DOWN, downward long-wave radiation, LR_UP: upward long-wave radiation, T_sensor: sensor temperature; Tair: air temperature, Wair: air wetness, Pair: air pressure, Win: wind speed.

4. Did the data undergo any QA/QC or are they posted 'as is'. Please discuss.

Re: The data is in situ observation. For the snow pit observation, multiple repeat observation was conducted to decrease the error. These statements were presented in the first two paragraphs in section 3.

L292-297: "The values from the three-time measurements for snow density in each layer were averaged to obtain the final snow density. The length of the longest and shortest axes of particles in each photo were measured using the software. The average lengths of longest and shortest axes from all photos in each layer were obtained as the final grain size."

The gap and abnormal values in the time series of automated layered snow temperature and 4-component radiation data were firstly replaced by Nan, and then were consolidated into a NetCDF file. The weather and soil data requested from ANRMS have been consolidated by ANRMS. The brightness temperatures, and weather and soil data requested from ANRMS were provided "as is".

L298-305: "The time series of automated layered snow temperature and 4-component radiation data were firstly processed with removal of abnormal values and gap fill, and then were consolidated into a NetCDF file "ten-minute 4 component radiation and snow temperature.nc". The ground-based brightness temperatures and the formatted weather and soil data requested from ANRMS were provided 'as is'. Brightness temperature data were divided into time series of brightness temperature and multi-angle brightness temperatures, and separately stored in two NetCDF file, and the weather and soil data were consolidated into a NetCDF file "hourly meteorological and soil data.nc"."

5. Please provide instrument prevision and accuracy information where possible. This information could be included in Table 2.

Re: Thanks for the recommendation. The accuracy of microwave radiometer was introduced in table 1. Because section 2.2 and 2.3 were merged, the table 2 was divided into two tables. The instrument precision of the instrument for snow pit observation were presented in table 3, and those for automatic observation were included in table 4.

Table 3. Variables collected by manual daily snow pit measurement in black field in figure 1, and their observation instruments, observation time and frequencies.

| Parameter | Instruments | Precision | Layering style | Observation time or frequency | Absent date |
|---|---|---|---|---|---|
| Layer thickness (cm) | Ruler | 0.1cm | Natural layering | local time 8:00-10:00 am | no |
| Snow density (g/cm$^3$) | Snow tube (Chinese Meteorological administration) | pressure:0. 1g/cm$^2$, snow depth: 0.1 cm | Whole snowpack | | no |

| Snow density (g/cm$^3$) | Snow shovel (NIEER) | weight: 0.01g, volume: 1cm$^3$ | Every 10 cm | |
|---|---|---|---|---|
| Snow density (g/cm$^3$) and | Snow fork (Toikka Enginnering Ltd.) | 0.0001g/cm$^3$ | Every 5 cm | January 2-3, 2016; February 20, 2016 |
| Liquid water content (%) | Snow fork | 0.001% | Every 5 cm | |
| Snow grain size (mm) | Anyty V500IR/UV | 0.001mm | Natural layering | December 24, 31, 2015; |
| Snow grain shape | Shape card | no | Natural layering | January 1-3, 23, 2016, February 20, 2016 |

**Table 4. Automatically observed variables and the observation instruments, observation time and frequencies.**

| Parameter | Instruments | Precision | Layering style | Observation time or frequency |
|---|---|---|---|---|
| Snow temperature(°C) | Temperature sensors (Campbell 109S) | 0.001 °C | 0 cm, 5 cm, 10 cm, 15 cm, 25 cm, 35 cm, 45 cm, and 55 cm | Ten-minute |
| 4-component radiation (W/m$^2$) | Component Net Radiometer NR01 (Hukseflux) | 0.001 W/m$^2$ | 6 feets above ground | Ten-minute |
| Soil temperature (°C) | Soil temperature sensor (China Huayun) | 0.1 °C | -5cm, -10 cm, -15cm and -20 cm | Hourly |
| Soil moisture (%) | Soil moisture sensor (DZN3, China Huayun) | 0.1% | -10 cm and -20 cm | Hourly |
| Air temperature (°C) | Thermometer screen (China Huayun) | 0.1 °C | 6 feet above ground | Hourly |
| Air pressure (hPa) | Thermometer screen (China Huayun) | 0.1 hPa | 6 feet above ground | Hourly |
| Air humidity (%) | Thermometer screen(China Huayun) | 1% | 6 feet above ground | Hourly |
| Wind speed (m/s) | Wind sensor(China Huayun) | 0.1m/s | 10 m above ground | Hourly |

**Manuscript consistency**

- Check for consistent use of upper and lower case throughout.

- Use consistent units for air temperature.

- Suggest 'entire snow season' or 'full snow season' instead of 'whole snow season' throughout

- Suggest 'snow layer' instead of 'layering snow' throughout

Re: Thank you very much for these detail problem. We checked all upper and lower case, names and units to make sure the consistency throughout the manuscript.

The air temperature unit in figure 12 were changed to $^oC$.

"layering snow" were replaced by "layered snow"

"full snow season" instead of "whole snow season"

You state that measurements of meteorological and soil parameters were requested from the ANRMS. Why these measurements were requested? In the context of your experiment, why is it important to have these data, in combination with the measurements of snow physical characteristics and microwave data? Please state in manuscript.

Re: Thanks for the suggestion. We added the function of environment data in the analysis of snowpack microwave emission transfer process.

In section 2.2 measurement methods:

A sentence was added:

L158: "The microwave signatures from snowpack vary with snow characteristics, soil and weather conditions."

In section discussion, we also presented

L487-489: "Snow grain sizes and snow densities within different layers presented different growth rates at during different temporal phasetime periods. Generally, the growth rates are related to the air temperature, pressure and snow depth (Chen et al., 2020; Essery, 2015; Vionnet et al., 2012; Lehning et al., 2002);"

So the meteorological and soil data are important in the microwave transfer process of snowpack, and snow characteristic evolution process.

With three difference snow density measurements can you provide any guidance on which ones might be most appropriate for different applications?

Re: We compared the three snow density, and found the results from snow shovel and snow tube are highly consistence. The results from snow fork are lower than the other two. The snow shovel and snow tube are traditional weighting methods. The measurements from these two instruments are more accurate than snow fork, but snow fork can get more precise vertical profile of snow density.

So, if studies need precise vertical profile of snow density, such as developing models, the data from snow fork will be given priority, but the data should be calibrated by weighting method.   If we need density to calculate SWE, the snow tube data is enough.

Because the comparison result was described in Dai et al. (2020), we did not present in this manuscript.

[Figure]

So we added explanation:

L364-366: Snow fork provided most detail snow density profile, but it systematically underestimated snow density compared with snow tube and snow shovel by 24% (Dai et al., 2022).

Besides, in snow melt period, large liquid water content would influence the measurement results of snow fork.

In section 5.2, we added the following explanation:

L504-506: In snow melt period, large liquid water content would influence the measurement results of snow fork. So, it is suggested to use small-size snow shovel or cutter to observe layered snow density in future experiments.

The phrase 'the collected data in this study include ground-based brightness temperatures, 4-component radiation, snow pit data, meteorological data and

automatically observed layering snow and soil temperatures.' or similar repeats multiple times. Could use less frequently to shorten the text and improve readability.

Re: this sentence was replaced by

L147-148: "Overall, the experiment performed a systematic observation covering electromagnetic and physical features of snow pack, providing data for studies on snow remote sensing and models."

**Minor line items**

L24-26: Sentence not clear. Do you mean 'evolution' processes?

Re: Yes, it is corrected.

L25: suggest either 'for evaluating' or 'to evaluated and improve'

Re: It is revised.

L77: do you mean evolution of snow parameters? Unclear.

Re: It is revised.

L85: Longer time series of data compared to what? Unclear.

Re: "the NoSREx and Japan radiation experiments were of fixed field observation, which provided longer time series of data." was revised to

L85: "The NoSREx and Japan radiation experiments were of fixed field observation, which provided longer time series of data than CLPX and SnowEx."

L204: To make it even more clear that a new snow pit was dug each day suggest writing 'In the black field, a new snow pit was dug each day.' This is an important part of your experiment so want to make it absolutely clear.

Re: Thanks, it was revised as you suggested.

L207-211: Nice. Thank you for this description and detail.

Re: Thanks.

L216: What was the constant interval of the snow density measurements?

Re: It was explained in the snow density measurement. The interval is 5 cm for snow fork measurement, and 10 cm for snow shovel.

L217: Please specify which software was used.

Re: the name of the software is "VIEWTER Plus", and we added in the text. Please see L220.

L246: for clarity suggest 'at 5 cm intervals starting 5 cm above the snow-soil interface

Re: It was revised.

Section 3 – When listing each dataset, please be consistent and include how each is stored. Also, maybe cross-reference with earlier sections, Tables and/or Figures.

Re: Thanks for the remind.

L293-295. Were the 17 samples at any sort of fixed frequency or just random dates?

Re: at the beginning, we conducted multi-angle observation after a snowfall. After Jan 3, snowpack continued densifying, and the observation was conducted every 5 days. After February, snow depth kept stable, only two times of multi-angle observations were conducted until March 3 when snowpack began to melt. After Mar 3, the observation frequency increased.

It was described in L185-189:

L184-188: Multi-angle observations were conducted after every big snowfall, and every 5 days in the stable period. In the melt period, observation frequency increased. There are total seventeen multi-angle observation (December 3, 19, and 30; January 3, 8, 13, 18, 3, and 28; February 3; March 3, 10, 15, 22, 26, 28, and 31) when the radiometer was set to scan the ground at different incidence angles at two ends of the orbit and the middle place of the orbit.

L288: Was any QZ/QC conducted? If not, perhaps add a sentence stating that the data are provided 'as is'.

Re: When collecting snow pit data, we conducted multiple times of observation to decrease the error. the grain size and snow density are the average value. The meteorological data requested from ANRMS had been undergone quality control. The automatic collected brightness temperature and snow temperature data are provided 'as is'.

Here we state it using below sentence:

L298-304: "The time series of automated layered snow temperature and 4-component radiation data were firstly processed with removal of abnormal values and gap fill, and then were consolidated into a NetCDF file "ten-minute 4 component radiation and snow temperature.nc". The ground-based brightness temperatures and the formatted weather and soil data requested from ANRMS were provided 'as is'. Brightness temperature data were divided into time series of brightness temperature and multi-angle brightness temperatures, and separately stored in two NetCDF files, and the weather and soil data were consolidated into a NetCDF file "hourly meteorological and soil data.nc".

L335: grain size of all fresh snow that fell during the 2015/2016 snow season or a specific event?

Re: Thanks. It is during 2015/2016. The sentence was revised to

"The grain size of the fresh snow was approximately 0.3 mm during the experiment."

L354: when did this 'stable phase' occur?

Re: after Jan 3, snow density kept stable, and slightly increase.

L362: "in the stable phase" was revised to "after January 3, 2016"

L365-366: I find these sentences rather confusing. It's not clear what you are trying to say.

Do you mean the diurnal range decreased from the top to bottom layers and as the snow depth increased there were more layers with diurnal temperature variations?

Re: Sorry for the confusing sentence. It means that the diurnal range decreased from the top to bottom layers and as the snow depth increased there were more layers with small diurnal temperature variations

"The diurnal variation range decreased from top to bottom layers, and with the increase of snow 366 depth, temperatures in more layers presented small diurnal variations"

Was revised to

L377-378: "The diurnal range decreased from top to bottom layers and as the snow depth increased there were more layers with small diurnal variations."

L387-390 (4.3 Brightness temperature): Fig 11a shows the brightness temperatures continuing to increase after 15 Jan when the snow density became

stable (Fig 8). Any insight as to what might be causing this? What do the crystal sizes show?

Re: The brightness temperature was mainly controlled by snow depth and grain size, and snow density. Snow density is the smallest influence factor. After 15 Jan, SWE changed little, but grain size continued increasing, Brightness temperature decreased with increasing grain size, due to the volume scattering increase.

L448: which phenomena?

Re: "These phenomena must rely on the environmental conditions, snow characteristics and soil conditions." was revised to

L470-471: "The different polarization behavior at 18 and 36 GHz might be related to the environmental conditions, snow characteristics and soil conditions."

**Tables and Figures**

Table 2

- Given the scope and aim of the journal, please include instrument precision and accuracy where possible. Could add as column to Table 2.
- Snow tube (L238 lists Chinese Meteorological administration, add this to Table 2) and snow fork models and manufacturer? Are these also produced by China Huayun? Please list the NR01 manufacturer.

Re: Thanks, the precision of instrument and their manufacturer are supplemented in table. Considering other reviewers' suggestion, table 2 was divided into two tables. Please see table 3 and 4.

The NR01 manufacturer is Hukseflux.

Snow fork was manufactured by Toikka Engineering Ltd. A Finnish radio- and microwave technology company

Figure 1 caption: 'in Asia' (delete 'the'). Delete 'Note: The map in the up right corner is ArcGIS self-contained map.

Re: it was revised

Figure 4:

- Please spell out CNR4 in the caption as I don't think it is used elsewhere in the text.

Re: Sorry, it should be CNR01. It was revised.

- Consider annotating the figure and sub-figures. i.e. upper left is 4-component radiation sensor, right is the snow profile sensor, center is the primary meteorological station, etc.

Re: it was changed according to your suggestion.

[Figure]

Was revised to

[Figure]

Fig 9. Please clarify in the caption that 0 cm is the snow/soil interface.

Re: it was revised to 0 cm (snow/soil interface). Please see figure 10.

Figure 10

- caption: remove 'variation'

- specify in the caption that 5 cm is 5 cm below the surface

Re:
"Figure 10: Hourly variation in soil temperature at 5 cm,10 cm, 15 cm and 20 cm (a) , and soil moisture at 10 cm and 20 cm (b)."
Was revised to
"Figure 11: Hourly soil temperature at 5 cm, 10 cm, 15 cm and 20 cm below the snow/soil interface (a), and soil moisture at 10 cm and 20 cm below the snow/soil interface (b)."

Figure 11

- the pink lines in Fig 11a (TBDh and TBDv) are not in the legend
- there is no a or b on the figures

Re: TBD =Tb18-Tb36. We revised the caption.

Combining other reviewers' suggestion, it was divided into two figures.

[Figure]

.

[Figure]

**Figure 11: (a) Daily variations in brightness temperatures at 1.4 GHz, 18 GHz and 36 GHz, for horizontal (Tb1h, Tb18h, Tb36h) and vertical polarizations (Tb1v, Tb18v, Tb36v), and the differences between Tb18h and Tb36h (TBDh), and between Tb18v and Tb36v (TBDv), at 1:00 am (local time), from November 27, 2015 to March 26, 2016. (b) hourly variation in Tb1h, Tb18h, Tb36h, Tb1v, Tb18v, Tb36v, from February 1 to March 23, 2016.**

"

Was revised to

"

[Figure]

**Figure 12: Daily variations in brightness temperatures at 1.4 GHz, 18 GHz and 36 GHz, for horizontal (Tb1h, Tb18h, Tb36h) and vertical polarizations (Tb1v, Tb18v, Tb36v), and the differences between Tb18h and Tb36h (Tb18h - Tb36h, and between Tb18v and Tb36v (Tb18v - Tb36v), at 1:00 am (local time), from November 27, 2015 to March 26, 2016. (a)for horizontal polarization, and (b) for vertical polarization.**

[Figure]

Figure 13 Hourly variation in Tb1h, Tb18h, Tb36h, Tb1v, Tb18v, and Tb36v (a), air temperature, soil moisture at 10 cm and soil temperature at 5 cm, and daily variation in snow depth (b), from February 1 to March 28, 2016."

**Additional minor editorial suggestions**

Please consider these minor editorial suggestions. These suggestions are not exhaustive. Consider additional proofing beyond what is listed here.

Thank you very much for the details.

L25-26: suggest '**to evaluate and improve** snow depth and SWE …

Re: it was revised.

L36: suggest produced or developed instead of 'was achieved'

Re: it was not accepted. The data was developed by NIEER, and released in the national Tibetan Plateau Data Center.

L37: '**at**' instead of 'in'

Re: it was revised.

L37-39: suggested revision: 'This unique dataset includes continuous daily snow pit data and coincident microwave brightness temperatures, radiation, and meteorological data, at a fixed site over a full snow season.'

"The dataset is unique in providing continuous daily snow pits data over a snow season at a fixed site and matched microwave brightness temperatures, radiation and meteorological data."

Was revised to

L37-38: "The dataset is unique in providing continuous daily snow pits data and coincident microwave brightness temperatures, radiation and meteorological data, at a fixed site over a full season"

L39-40: is expected to serve the evaluation and development of microwave

Re: it was revised.

L48: 'processes'

Re: it was revised.

L 49: '**is** controlled'

Re: it was revised.

L 50: 'and variations in snow characteristics cause uncertainties in albedo estimation' (drop 'the' x2)

Re: it was revised.

L51: '**at** global and regional scales'

Re: it was revised.

L56: '**of** electromagnetic and … improve understanding **of** the…'

Re: it was revised.

L61: **to** produce

Re: it was revised.

L62: **have** been

Re: it was revised.

L63-64: The Cold Land Processes Field Experiment (CLPX) (https://nsidc.org/data/clpx/index.html), one of the most well-known experiments, was carried out from winter of 2002 to spring of 2003 in Colorado, USA (Cline et al., 2003).

Re: it was revised.

L65-66: snow pits were collected **in** February and March of 2002 and 2003 to coincide with airborne…

Re: it was revised.

L67: **to test and develop** instead of 'develop/test'

Re: it was revised.

L70: In northern Canada, (drop 'region' and Canada instead of Canadian)

Re: it was revised

L76: pit observations **were** used **to** evaluate

Re: it was revised.

L81: boreal forest spanning an entire winter season

Re: it was revised.

L82: **at** 3 or 4 day intervals

Re: it was revised

L85: were fixed field (drop 'of')

Re: it was revised

L85-89: Unclear. Possible rephrasing – 'The experiments were conduced in a deep snow area and the weekly observation interval permits observation of general snow evolution characteristics but might miss some key details that occur at sub-weekly scales.' . Delete sentence starting with 'furthermore on L87-89.

"These experiments were conducted in deep snow area, and the week-interval observation could reflect the general evolution process of snow characteristics, but might miss some details. Furthermore, in the area with snow cover duration within 4 months, the week-interval observation hardly depicts the change details."

Was revised to

L87-89: "These experiments were conducted in deep snow areas, and the weekly- observation could reflect the general evolution process of snow characteristics, but might miss some key details that occur at sub-weekly scales."

L90: To understand the evolution of

Re: "comprehensively" was deleted.

L98: continuous

Re: it was revised

L100: location, parameters, **and** parameter measurement

Re: it was revised

L101: **at** the National

Re: it was revised

L103: the possible **applications**

Re: it was revised

L107-108: was performed during the 2015/2016 snow season

Re: it was revised

L110-111: which is approximately 6 km from the foot of the Altay mountain in northwest China (Figure 1).

Re: it was revised

L112: **provides** snow water resources for **these** four countries.

Re: it was revised

L114: 40 cm, with a maximum over 70 cm.

Re: it was revised

L117: with **areas** of

Re: it was revised

L120-121: was set up in the middle of the field, facing

Re: it was revised

L126: The blue field **was** (try to use consistent verb tenses)

Re: it was revised

L127: precipitation, **soil layer** temperature, **and** soil moisture

Re: it was revised

L129: observations **of** snow depth (or daily manual snow depth and SWE observations)

Re: it was revised

L132-136: This does not need to be its own paragraph. Append to end of previous one. L138: between **them is** less than 100 m

Re: it was revised

L157-158: The **automated** data collection frequency was set **to** 1 minute.

Re: it was revised.

L167: suggest 'the automated snow temperatures collected in the red field'

Table 2: '**feet'** instead of 'feets' (**6 feet above ground**)

Re: they are revised.

L171-172: The soil and weather parameters are routine observations conducted at

Re: it was revised

L179-182: Suggest deleting. This same text repeats often and is not necessary here.

Re: section 2.2 and 2.3 were merged. This paragraph was kept, and "the prior same text" was deleted.

L184: Before **the** (or Prior to the)

Re: "the" was added.

L186: in clear sky conditions (delete 'the')

Re: it was revised

L190: fixed in the middle of the orbit (delete 'place')

Re: it was revised

L196-198: Therefore, the snow are snow characteristics were considered homogeneous within the field of view of the antennas.

Re:

"Therefore, the snow and soil characteristics presented homogeneous distribution within the view field of the three antennas."

Was revised to

L191-192: "Therefore, the snow and soil characteristics were considered homogeneous within the view field of the three antennas."

L202: snow layering, layer grain size and type, and snow layer density.

Re: it was revised to

L198-199: "snow layering, snow layer thickness, snow grain size and type, snow density, and snow temperature"

L203: making **a** snow pit

Re: it was revised

L206: delete 'for observers to conveniently observe.'

Re: it was revised

L207: snow profile **is exposed**

Re: it was revised

L212-214: Combine with previous paragraph. Could also rephrase to: '...the natural snowpack stratification was visually determined and the thickness of each layer measured using a ruler.'

Re: this paragraph was revised as below, and combined with previous paragraph.

L216-217: "After finishing a snow pit, the natural snowpack stratification was then visually determined, and the thickness of each layer was measured using a ruler."

L217-218: with an 'Anyty V500IR/UV' camera (Figure 2a)

Re: it was revised

L221: In this experiment, a ruler with 0.5 mm marking was used as a reference

Re:

"In this experiment, the minimum scale "0.5 mm" of a ruler was used as reference"

Was revised to

L223-224: "In this experiment, a ruler with 0.5 mm marking was used as a reference"

L227-228: Each layer had at least 10 groups of longest and shortest axes length; the final grain size was the average of these values.

Re: it was replaced by the sentence you advised.

"in each layer, there were at least 10 groups of the longest and shortest axes length were obtained, and the final grain size was the average values."

Was revised to

L230-231: "Each layer had at least 10 groups of longest and shortest axes length; the final grain size was the average of these values."

L247: Table A2 is an example record table for snow density.

Re: it was revised

L247-248: Three observations were conducted for each layer.

Re: it was revised

L255: suggest 'Snow layer temperatures were collected using temperature sensors in the red field instead of …'

Re: due to the merge of section 2.2. and 2.3, this sentence was deleted.

L256: sensors **were** set up

Re: "had been" was revised to "were"

L258: , and 55 cm from the base of the soil-snow interface.

Re: it was revised to "55 cm above soil/snow interface.

L259: typo – need space between The and NR01

Re: it was corrected.

L260: far infrared

Re: "Far Infrared" was revised to "far infrared"

L282: reorganized **and** consolidated **for ease of use**.

Re: "or" was changed to "and". "easily usage" was revised to "ease of use".

L295: angle, **and** brightness temperatures

Re: "and" was added.

L314: **is** described

Re: "were" was corrected to "is"

L327-331: awkward text. Suggest rephrasing. Possible change: 'During this snow season there were seven snowfall events, each formed a distinct snow layer except for the third event whose layering became indistinguishable from the second layer (Figure 6 gray). The fourth event was the biggest, after which time the snow depth started to decrease and snow density increased.'

Re: "During this snow cover duration, seven snowfall events occurred, and each snowfall formed one layer in snow cover on the ground, except the third event which presented a new layer on the second layer at the beginning, but the layering interface disappeared after several days and visually displayed as one layer (in gray in Figure 6). The fourth event was biggest of all, and the depth of snow cover exhibited decreasing with increase of snow density after the fourth event. Snow cover began melting on March 14 and snow depth declined to zero within 10 days."

 Was revised to

L337-339: "During this snow season, there were seven snowfall events, each formed a distinct snow layer except for the third event whose layering became indistinguishable from the second layer (Figure 7 gray). The fourth event was the

biggest, after which time the snow depth started to decrease and snow density increased."

L333: within all layers increased during the

Re: it was revised.

L335: perhaps top to bottom instead of up to down? And for other occurrences of this phrasing.

Re: all "up to down" was replaced by "top to bottom".

L336-337: The biggest long are short axes were 6 cm and 4 cm, respectively, and occurred in Layer 1 during the melt period.

Re: "which occurred within Layer 1 in the melting period. " was revised to "and occurred in Layer1 during the melt period"

L339: above 0°C ?? not clear

Re: the unit °C was added.

L339-340: snowpack melt accelerated

Re: it was revised

L350: suggest 'instruments' instead of 'equipment'

Re: all 'equiment' were replaced by 'instruments' through the paper.

L351: and remained stable after reaching ~0.2-0.25 g/cm3.

Re: it was revised

L352-535: From March 14 on, snow densities abruptly increased and the maximum value reached was over 0.45g/cm3.

Re: "From March 14 on, snow densities abruptly increased. The biggest value was beyond 0.45g/cm3." was revised to

L360: "From March 14 on, snow densities abruptly increased, and the maximum value reached was over 0.45g/cm$^3$."

L364: Snow temperature in the top layer had the largest diurnal variation.

Re: it was revised.

L371-372: remained stable and below 0°C during the snow season but had large fluctuations before and after snow on/off.

"The soil temperature at 5 and 10 cm remained below 0 °C and stable during the snow season, but presented large fluctuation before snow cover onset and after snow off" was revised to

L384-385: "the soil temperature at 5 and 10 cm remained stable and below 0 °C during the snow season but presented large fluctuation before (after) snow on (off)"

L373: do you mean temperature difference? Suggest: 'The temperature difference between 5 cm and 10 cm was much larger before snow cover onset than during the snow cover period.

"The temperature gaps between 5 cm and 10 cm were much larger before snow cover onset than those during snow cover duration." Was revised to

L385-386: "The temperature difference between 5 cm and 10 cm was much larger before snow cover onset than during snow cover period."

L375: suggest 'snow cover period'

Re: "snow cover duration" was changed to snow cover period

L376: I found this sentence a little confusing. Suggest 'Within the snow cover period, there were two soil moisture peaks, one from 12-14 Dec and another from 1-20 Jan.

"The soil moistures at 10 cm were above 10% before snow cover onset and after snow off, and were below 10% during the snow cover duration. During Dec 12-14, and Jan 1- 20, soil moisture showed peak value, which corresponded to the two high-value periods of soil temperature."

was revised to

L386-388: "The soil moistures at 10 cm were above 10% before snow cover onset and after snow off, and were below 10%, and there were two soil moisture peaks, one from December 12-14 and another from January 1- 20, within the snow cover period."

L384: 1:00 am local time?

Re: Yes, "local time" was added

L384. Suggest starting a new sentence with 'Figure 11b'

Re: it was revised.

L385: Maybe list the three frequencies in parentheses to remind the reader.

Re: "at the three frequencies" was revised to " at 1.4, 18 and 36GHz".

L386: Data **show**

Re: "depict" was revised to "show"

L386-387: Tb18h shows an obvious decline after Feb 18, and Tb18v after Mar 3 (Figure 11a).

Re: "Tb18h show obvious decline after Feb 18, and Tb18v show decline after Mar 3 for vertical polarization"

was revised to
L399: "Tb18h shows an obvious decline after February 18, and Tb18v after March 3"

L389: snow density became stable on Jan 15.

Re: "arrived at" was revised to "became"

L391-392: suggest '…exhibited a distinct cycle of daytime increases and nighttime decreases, resulting from high daytime air temperatures (above 270K) and associated melt-freeze cycles.

Re: "After Feb 25, brightness temperature exhibited abrupt increase (at day time) - decrease (at night time) circle (Figure 11b), because air temperature at noon increasing up to above 270 K resulted in large liquid water content at day time, and the melted snowpack refroze when air temperature decreased at night time and brightness temperature decreased"

was revised to

L402-404: "After February 25, brightness temperature exhibited a distinct cycle of daytime increase and nighttime decrease (Figure 13), resulting from large liquid water content caused by high daytime air temperature (above 0°C) and the melted snowpack refreezing at nighttime."

L395-397: suggest 'After March 14 there was another big rise in air temperatures and even the nighttime air temperatures were above 270 K. During this period of accelerated snowmelt the liquid water within the snowpack did not refreeze completely at night and both the brightness temperature and brightness temperature difference showed irregular behaviour.'

Re: It was revised as you suggest. Please see L405-408.

"After March 14, there was another big rise in air temperature and even the nighttime air temperatures were above 0oC. During this period of accelerated snowmelt, the liquid water within the snowpack did not refreeze completely at night and both the brightness temperature and brightness temperature difference exhibited irregular behavior."

L412-413: suggest 'Although the magnitudes differ, the general temporal patterns are the same, even the abrupt change between 3 and 4 Mar is captured by both instruments. The correlation coefficients at both …'

Re: "Although there was large difference between them, the general variations are the same, even for the abrupt change between Mar 3 and Mar 4, and the correlation coefficients at"

Was revised to

L425-428: "Although there were large differences between satellite and ground-based observations, the general temporal patterns are the same, even the abrupt change between March 3 and March 4 is captured by both satellite and ground-based sensors. The correlation coefficients at"

L423: **downward** short-wave

Re: it was revised

L426-427: Can you put the snow on and off dates in parentheses?

Re: They were added.

L441-442: The upward short-wave radiation abruptly increased when the ground was covered by snow (after November 21), and sharply declined on the snow off day (March 25).

L428: **by** the end of the snow season

Re: "to"-"by"

L423: models

Re: is it L433?

 "updating microwave emission transfer model of snowpack" was revised to

L457: "updating a microwave emission transfer model of snowpack"

L437: the dominant control (delete 'factor')

Re: it was deleted.

L438: **did** not correspond

Re: it was revised.

L439: do you mean brightness temperature **difference** of the dry snowpack?

Re: yes, brightness temperature difference between 18 and 36 GHz.

It was revised.

L440: do you mean maximum difference (instead of gradient)?

Re: maximum difference replaced the peak gradient

L443: had similar **variations**

Re: It was revised.

L444: **time** periods

Re: It was revised.

L445: and was less stable

Re: It was revised.

L449: were absent **from Dai et al (2021)'s** simulation so the dynamic ground…

Re: sorry for the confusion. It means the subsurface (within 5 cm) soil moisture was not observed.

"the subsurface soil wetness data were absent" was revised to

L472: "the subsurface soil moisture was not observed"

L450: **predominantly** instead of dominantly

Re: It was revised.

L451: conditions

Re: It was revised.

L455: influences

Re: It was revised.

L456: **the** climate system

Re: "the" was added.

L456-458: The factors altering snow surface albedo are

Re: It was revised.

L459: ,while others considered snow albedo to depend mainly on snow aging.

Re: It was revised.

L463: albedo **models**

Re: It was revised.

L465-466: within different layers **had** different growth rates **during** different **time periods**

"Snow grain sizes and snow densities within different layers presented different growth rates at    different temporal phase"

Was revised to

L487: "Snow grain sizes and snow densities within different layers presented different growth rates during different time periods."

L473: at **a** fixed site

Re: "the" was revised "a"

L474: which provide **a detailed** description of

Re: it was revised

L480-482: delete 'Actually'.

Re: it was deleted.

L474-487: combine into a single paragraph.

Re: it was done.

L484: 'Existing studies report that …

Re: "The existing studies" was revised to "existing studies"

L486: These data provide a good opportunity to

Re : "It is a good chance to analyze" was revised to
L534:" These data provide a good opportunity to".

---

## Author Comment (AC3)

In this manuscript, a set of long time series microwave radiation snow observation experiment data obtained in Altay region of Xinjiang during the 2015/2016 snow season were described and discussed in detail, including the test area overview, measurement methods, data arrangement and preliminary result analysis of measurements. This is a very comprehensive and unique measured dataset, these datasets including: microwave brightness temperature data with dual polarization in three bands, snow characteristics data, four-component radiation observation data and meteorological observation data, etc. According to the preliminary result analysis of the measured data, this set of data has very high measurement quality, which is of great value for the better input of snow model development, the verification of simulation results and related snow application. The full text is written in standard English, logical and fluent, with good readability.

Thanks for these constructive suggestions. We have revised the manuscript according to your comments. Please see following point-to-point response.

However, there are the following related issues need to clarify or modify:

1. The standards or specifications for this experimental implementation can be supplemented;

Re: Thanks for the suggestion. Another reviewer also pointed out unclear description of the experiment, and suggest merging section 2.2 and 2.3. based on your suggestions, we reorganized section 2.2 and 2.3. In measurement of every parameter, we described the instrument, instrument precision, observation frequency, time and observation protocols.

We do not list out all change. Please see detailed revision in "section 2.2 measurement methods" in the revised manuscript.

2. Please describe and supplement the measurement error analysis and data quality control method of the datasets in detail;

Re: The data is in situ observation. For the snow pit observation, multiple repeat observation was conducted to decrease the error. For example, the snow density observation was conducted three times. If one of the three value is abnormal, the fourth observation would be performed. The final density is the average value of the three normal values. For the automated observation, the temporal curves were drawn to remove the abnormal values. However, we believe there is more or less errors caused by instruments or manual operations. Moreover, we adopted another reviewer's suggestion to consolidate data into NetCDF files.

1) In section 3 Description of released IMCS data

"The ground-based brightness temperatures layered snow temperatures, and 4-component radiation were automatically collected in uniform format" was revised to

L298-305: "The time series of automated layered snow temperature and 4-component radiation data were firstly processed with removal of abnormal values and gap fill, and then were consolidated into a NetCDF file "ten-minute 4 component radiation and snow temperature.nc". The ground-based brightness temperatures and the formatted weather and soil data requested from ANRMS were provided 'as is'. Brightness temperature data were divided into time series of brightness temperature and multi-angle brightness temperatures, and separately stored in two NetCDF files, and the weather and soil data were consolidated into a NetCDF file "hourly meteorological and soil data.nc". Table 3 describes the contents of the provided dataset."

In section 5 Discussion, we added the uncertainties of the dataset.

2) we added a paragraph in section 5.1:

L450-455: Although the dataset is just for one season observation, the daily snow pit observation with coincident microwave and optical radiation data in a full snow season provide the most detailed variation of snow parameters which allow researchers to find more details in snow characteristics and their relationship with remote sensing signatures. The dataset also fills the snow observation gap in mid-low snow depth area with relative short snow cover duration.

3) we added section 5.2 Uncertainties to present the short observation of L band, no subsurface soil moisture, and the subjectivity of grain size measurement. (L492-509)

5.2 Uncertainties

During the experiment, some uncertainties were produced due to irresistible factors. It is reported that the sampling depth of the L-band microwave emission under frozen and thawed soil conditions is determined at 2.5 cm (Zheng et al., 2019). We did not collect subsurface soil moisture, and the L band radiometer observation began on January 30, 2016. Therefore, it is difficult to obtain the ground emissivity in the full snow season based on the data. The soil moisture data at 10 and 20 cm under soil/snow interface cannot be directly used to validate and develop soil moisture retrieval from L band brightness temperature. We hope detailed soil moisture profile will be observed to estimate the subsurface soil moisture to fill the gap.

The grain size data were collected through taking photos. When measuring the length of grains, the grain selection has subjectivity, and the released data are average values. Although the general variation trend can be reflected by the

time series of average grain size, some details might be missed. Therefore, the original grain photos could be provided through requesting for authors. In snow melt period, large liquid water content would influence the measurement results of snow fork. So, it is suggested to use small-size snow shovel or cutter to observe layered snow density in future experiments.

One season observation is quite valuable for developing and validate remote sensing method or snow model, although the representativeness of this observation remains unknown. We need more years of observation to endorse or confirm the evolution of snow characteristics.

3. If possible, some practical application cases study related to this dataset can be added.

Re: Thanks for this suggestion.

We published a paper "Improving the snow volume scattering algorithm in a microwave forward model by using ground-based remote sensing snow observations" in "IEEE Transactions on Geoscience and Remote Sensing", based on this dataset. We used the data to develop a new volume scatter algorithm. The reviewers encouraged us to publish a summary paper to describe this dataset. So we did not describe specific cases, but described the existing application and discussed the possible applications.

In section "5.1 Discussion", we presented existing and possible applications:

"5.1 Applications

Although the dataset is just for one season observation, the daily snow pit observation with coincident microwave and optical radiation data in a full snow season provide the most detailed variation of snow parameters which allow researchers to find more details in snow characteristics and their relationship with remote sensing signatures. The dataset also fills the snow observation gap in mid-low snow depth area with relative short snow cover duration.

The snow pit data and microwave brightness temperatures have proven useful for evaluating and updating a microwave emission transfer model of snowpack (Dai et al., 2022). This dataset reflected the general fact that brightness temperature at higher frequencies presented stronger volume scattering of snow grains, and were more sensitive to snow characteristics. This experiment revealed that the dominant control for the variation of brightness temperature was the variation of grain size but not the snow depth. The largest snow depth or SWE did not correspond to the largest brightness temperature difference between 18 and 36 GHz in the condition of dry snowpack. Due to the

growth of grain size, the maximum difference occurred before melting for stable snow cover. Therefore, the daily snow depth variations curve derived from passive microwave remote sensing datasets tend to exhibit a temporal offset from those of in situ observation.

During the snow season, brightness temperatures for both polarizations presented similar variations, but they behaved different in some time periods. The horizontal polarization was more sensitive to environment and was less stable than vertical polarization. Besides, the polarization difference at 18 GHz and 36 GHz showed increase and decrease trends, respectively during the experimental period. The results for 18 GHz were opposite to the simulation results (Dai et al., 2022). The different polarization behavior at 18 and 36 GHz might be related to the environmental conditions, snow characteristics and soil conditions. However, the subsurface soil moisture was not observed, the dynamic ground emissivity could not be estimated. L band has strong penetrability, and the brightness temperature variations were predominantly related to subsurface soil conditions, except when the liquid water content within snowpack was high. Therefore, in the condition of soil moisture data absence, L band brightness temperatures were expected to reflect soil moisture variation which influence the soil transmissivity (Babaeian et al., 2019; Naderpour et al., 2017; Hirahara et al., 2020).

Snow surface albedo significantly influences the incoming solar radiation, playing an important role in the climate system. The factors altering snow surface albedo contains the snow characteristics (grain size, SWE, liquid water content, impurities, surface temperature etc), external atmospheric condition and solar zenith angle (Aoki et al., 2003). Snow albedo was estimated based on snow surface temperatures in some models (Roesch et al., 1999), while others considered snow surface albedo to depend mainly on snow aging (Mabuchi et al., 1997). In this experiment, we obtained the 4-component radiation, snow pit and meteorological data. These data provide nearly all observations of possible influence factors, and could be utilized to discuss and analyze shortwave radiation process of snowpack, and validate or improve multiple-snow-layer albedo models.

Snow grain sizes and snow densities within different layers presented different growth rates during different time periods. Generally, the growth rates are related to the air temperature, pressure and snow depth (Chen et al., 2020; Essery, 2015; Vionnet et al., 2012; Lehning et al., 2002); therefore, this dataset can be used to analyze the evolution process of snow characteristics, as well as validation data for snow models."

4. Page 1: "Involution Processes" should it be "evolution processes"?

Re: We corrected it.

5. The literature of Dai et al., 2021 is mentioned in the paper, but only the literature of Dai et al., 2020 is found in the reference at the end of the paper, please check.

Re: We added the reference. Dai et a., 2021 was revised to Dai et al., 2022

Dai, L., Che, T., Xiao, L., Akynbekkyzy, M., Zhao, K., and Leppanen, L.: Improving the Snow Volume Scattering Algorithm in a Microwave Forward Model by Using Ground-Based Remote Sensing Snow Observations. IEEE Transactions on Geoscience and Remote Sensing, 60: 4300617. doi:10.1109/TGRS.2021.3064309, 2022.

6. In Line 186-189, there are two brightness temperature values in the two polarization of 1.4ghz, and only one value in the two polarization of the other two bands. Please check whether they are correct.

Re: For 18 and 36 GHz, the sky brightness temperatures for vertical and horizontal polarizations were close to each other.

To make the meaning clear, we revised this sentence:

"those were approximately 29.7$\pm$0.3 K and 29.3$\pm$0.9 K at 18.7GHz and 36.5 GHz, respectively."

was revised to

L179-180: "the sky brightness temperatures were approximately 29.7±0.3 K at 18.7 GHz for both polarizations and 29.3±0.9 K at 36.5 GHz for both polarizations."

7. Section 4 "Overview of collected Data from IMCS", could it be "Overview and preliminary analysis of collected data from IMCS"?

Re: We revised the title according your suggestion.

4 Overview and preliminary analysis of collected data from IMCS

Based on above comments, this manuscript can be accepted after minor revision.

---

## Author Comment (AC4)

Snow is a very important climate parameter. Any attempt to obtain snow physical properties, mass balance, as well as thermodynamic regimes deserve encouragement and promotion. For this reason, I support authors to publish high-quality datasets which have high potential to benefit earth climate research (ECR).   On the other hand, as a reviewer, I would rather tell constructive comments and even criticism, the goal is to improve the quality of the manuscript. For this reason, I have the following comments:

1) I find this manuscript didn't provide the best reading experience. Initially, I printed the manuscript and planned to read it while adding my notes on it, but eventually, I have to abandon such a plan because the text is too small to see comfortably. The authors are strongly suggested using at least 12 font sizes for the revised manuscript.

Re: Sorry for the bad reading experience. Although it is the journal template, we changed the font size.

2) The weakest point of this manuscript is that the authors presented data that has lasted for one season. It would be a much stronger data paper if observations covers multi-seasonal or even multi-decadal scales. On the other hand, one season may also represent a lot of useful information. So, I would like to see some explicit arguments to support just one single seasonal data presentation.

Re: Thanks for the suggestion.

We admit that the dataset is just for one season observation, nevertheless, the daily snow pit observation with coincident microwave and optical radiation data in a full snow season provide the most detailed variation of snow parameters which allow researchers to find more details in snow characteristics and their relationship with remote sensing signatures. The dataset also supplemented the snow observation gap in mid-low snow depth area with relative short snow cover duration. Actually, this dataset has been used to develop snow volume scatter algorithm.

In section 5.1 application, we added a paragraph:

L 451-455: "Although the dataset is just for one season observation, nevertheless, the daily snow pit observation with coincident microwave and optical radiation data in a full snow season provide the most detailed variation of snow parameters which allow researchers to find more details in snow characteristics and their relationship with remote sensing signatures. The dataset also fills the snow observation gap in mid-low snow depth area with relative short snow cover duration."

However, some limits exist. We added section 5.2 Uncertainties to present the limits.

5.2 Uncertainties (L493-509)

During the experiment, some uncertainties were produced due to irresistible factors. It is reported that the sampling depth of the L-band microwave emission under frozen and thawed soil conditions is determined at 2.5 cm (Zheng et al., 2019). We did not collect subsurface soil moisture, and the L band radiometer observation began on January 30, 2016. Therefore, it is difficult to obtain the ground emissivity in the full snow season based on the data. The soil moisture data at 10 and 20 cm under soil/snow interface cannot be directly used to validate and develop soil moisture retrieval from L band brightness temperature. We hope detailed soil moisture profile will be observed to estimate the subsurface soil moisture to fill the gap.

The grain size data were collected through taking photos. When measuring the length of grains, the grain selection has subjectivity, and the released data are average values. Although the general variation trend can be reflected by the time series of average grain size, some details might be missed. Therefore, the original grain photos could be provided through requesting for authors. In snow melt period, large liquid water content would influence the measurement results of snow fork. So, it is suggested to use small-size snow shovel or cutter to observe layered snow density in future experiments.

One season continuous observation is quite valuable for developing and validate remote sensing method or snow model, although the representativeness of this observation remains unknown. We need more years of observation to endorse or confirm the evolution of snow characteristics.

3) The manuscript is too long, there are a lot of technical details/specifications. I suggest authors move those materials to the appendix. In the main body of the manuscript, authors should mainly focus on descriptions of the observation and illustrations of the results.

Re: Thanks for the suggestion.

Because this is a data paper, other reviewers pay much attention on the technical specifications, and suggested to add the precision of instruments, and standards or specifications for this experimental implementation.

Therefore, combining four reviewers' suggestions, we kept the technical specifications in the main body of the manuscript, and merged 2.2 and 2.3 to shorten the manuscript.

Please see details in section 2.2.

4) The quality of most photos are not good (maybe because those are too small) and many of them are not very informative. I suggest the authors drop most of the photos but enlarge/enhance the size/quality of a few selected ones. The criteria of photo selections should be such that it either helps the readers to understand better the observation site (domain map and landscape) or the data you have collected (snowpits sites). Otherwise, the paper is more like a technical report.

Re: Thanks for the suggestion. We enlarged the photos of grains (Figure A1), the microwave radiometer was added as figure 2 in revised manuscript.

The equipment used for snow pit observation in figure 1 are enlarged and presented in figure 3 and figure 4. The equipment used for automatic observation in figure 1 are enlarged and presented in figure 5 and figure 6.

5) The time series of the figures and some tables can still be improved:

1. a) You have observed, e.g. snow density (fig.8b) at different levels. would it be possible to make a contour plot to show the layering effect of snow density? I believe such a contour plot can offer readers a lot more information to better understand the snow density spatiotemporal variations. The same effect can also apply to figure 9.

Re: Thanks for the suggestion. We tried to use contour plot to express the multi-layer snow density variation, but the result looks a little bit confusion (figure r1). So we used image to show the variation of snow density in all layers (figure r2) .

[Figure]

Figure r1 Snow density in contour style.

[Figure]

Figure r2 Snow density in image style

For the figure 8(c), there are 4 layers of snow density. The image style is not suitable to describe them. So, we changed the line color to make them more distinguishable.

[Figure]

2.  b) The x-labels of figure 9, 10, 11, 13 are not good, such high resolution and precise timestamp does not help much to understand the time series, e.g. the lines in figure 9 and 13 are so close to each other and we can hardly to see anything clear. I suggest authors use standard time-label, such as "day"

Re: Thanks, we revised the timestamp, and combining other reviewers's suggestion, figure 9, 10, 11, and 13 were revised as below:

figure 9:

[Figure]

Was revised to

[Figure]

Figure 10: Minutely variation in layered snow temperatures at 0 cm (snow/soil interface), 5 cm, 15 cm, 25 cm, 35 cm, 45 cm and 55 cm above ground during experiment time.

Figure 10:

[Figure]

Was revised to

[Figure]

Figure 11: Hourly soil temperature at 5 cm, 10 cm, 15 cm and 20 cm below the snow/soil interface (a), and soil moisture at 10 cm and 20 cm below the snow/soil interface (b).

Figure 11a

1. Figure 11a was divided into two figure. One for H polarization, another for V polarization.

[Figure]

Figure 12: Daily variations in brightness temperatures at 1.4 GHz, 18 GHz and 36 GHz, for horizontal (Tb1h, Tb18h, Tb36h) and vertical polarizations (Tb1v, Tb18v, Tb36v), and the differences between Tb18h and Tb36h (Tb18h - Tb36h, and between Tb18v and Tb36v (Tb18v - Tb36v), at 1:00 am (local time), from November 27, 2015 to March 26, 2016. (a)for horizontal polarization, and (b) for vertical polarization.

2. Figure 11a shows the brightness temperature through the whole snow season. Figure 11b focus on the melting phase. According to the comments, we added the variation in snow depth, soil moisture and soil temperature to link the variation in different parameters.

Figure 11b

[Figure]

was revised to:

[Figure]

Figure 13 Hourly variation in Tb1h, Tb18h, Tb36h, Tb1v, Tb18v, and Tb36v (a), air temperature, soil moisture at 10 cm and soil temperature at 5 cm, and daily variation in snow depth (b), from February 1 to March 28, 2016.

Figure 13:

[Figure]

was revised to

[Figure]

Figure 15: Minutely variation in 4-component radiation and daily variation in snow depth at Altay station from November 3 2015 to April 15 2016.

3. c) Figure A1 needs a major revision. you either enlarge the entire figure substantially or only show a selected example or drop this figure.

Re: We enlarged the entire figure to make sure it more clear. Besides, Figure 3 shows the example of grain photo.

[Figure]

**Figure A1: Photos of grains and reference ruler in each layer on February 15, 2016, and in each photo the longest and shortest axis lengths of the chosen grains are labeled.**

4.  d) Table A2: What do you expect readers to learn from this table? Or what is your message to the readers?

Re: This table presented the original record of snow density to explain measurement for each layer or method was conducted three times. It is just an example to show reader how we record the three kinds of snow density.

6) The conclusion is too short and too superficial.   Please give a quantitative summary of what you find from the snow observations. For example, what were the spatiotemporal variation of snow density (I see this is a very important data set and it deserves more attention), snow temperature regimes, and brightness temperature? Anything concluded from the data characteristics need to be described. It will boost the application and citation of your data sets.

Re: Thanks for the suggestion. Summary of data analysis results were added in conclusion.

L 516-520: "Generally, grain size grew with snow age, and increased from top to bottom. Snow grains are rounded shape with small grain size in the top layer, and depth hoar with large grain size in the bottom layer. Snow density experienced increase-stable-increase variation, and the densities of the middle layers were greater than the bottom layer due to the well-developed depth hoar in the stable period." was added in the second paragraph

L526-527: "Grain size is the most important factor to influence snow volume scattering." Was added in the third paragraph.

I hope authors may find my comments useful and helpful to make further improvements to their manuscript.

Re: Thank you very much for these constructive comments. They are truly helpful to improve our manuscript.

---

## Author Response (AR2)

Dear editor and reviewers,

Thanks for your comments, recommendations and revisions. Every comment and suggestion from reviewers have been responded point by point, and corresponding modifications and explanation had been reflected in the revised manuscript based on reviewers' suggestions. The summary of the changes includes:

1) We added a photo link in figure A1 to make reader clearly see the grain photos.
2) We merged the time variables year, month, day, hour, minute and second into a variable "time".
3) We asked for an English edition to make the paper more easily read, and checked all references and citations to make them consistent.

**Comments to the author**:

After carefully reviewing the attached reviews and your manuscript, I think this paper can be acceptable for publication, but it needs to be revised according to the reviewers' comments:

(1) Before final acceptance, I suggest that the authors still make the figure A1 presentation changes. I am not sure if ESSD can accept the digital presentation. It would be more informative to present those snow grain size photos on a much larger scale (say one photo per page) so readers can see them clearly. Or add the photo link in the figure caption so that readers can access the original high-quality images.

Re: Thanks for the recommendation. We uploaded all grain photos, and added a photo link in the caption. "Figure A1: Photos of grains and reference ruler in each layer on February 15, 2016, and in each photo the longest and shortest axis lengths of the chosen grains are labeled."

was revised to

Figure A1: Photos of grains and reference ruler in each layer on February 15, 2016. In each photo the longest and shortest axis lengths of the chosen grains are labeled. Original photos are in URL: http://arcticroute.tpdc.ac.cn/navigate/bmp

(2) It is very nice to combine several of the text files into NetCDF datasets. But the year/month/day/minute/second variables are redundant and can be removed because this information is contained in the 'time' coordinate.

Re: Thanks. We merged variables "year", "month", "day", "hour", "minute", "second" into a variable "time". For example: in "TBdata.nc", variables "year", "month", "day", "hour", "minute", "second" were merged as a variable "time" which was described as yyyy-mm-dd hh:mm:ss; in "Manual snow pit data.nc", variables year, month, day were merged as a variable "time" which was described as yyyy-mm-dd. Table 3 was also updated.

(3) The text is understandable but would benefit from English language editing.

Re: We asked for an English edition. Please see make-up manuscript.

(4) There are several in text citations missing from the reference list and referencing style is somewhat inconsistent.

Re: Thanks for the careful check. We checked the citations and references, removed the references not

cited in text, added the missing references, and completed all references according to journal guideline.

L52: Pulliainen et al., 2020

After English edition, Pulliainen et al., 2020 was not cited.

Moreover, we added:

Fierz, C., Armstrong, R.L., Durand, Y., Etchevers, P., Greene, E., McClung, D.M., Nishimura, K., Satyawali, P.K. and Sokratov, S.A.: The International Classification for Seasonal Snow on the Ground. IHP-VII Technical Documents in Hydrology N °83, IACS Contribution N °1, UNESCO-IHP, Paris, 2009.

In-text citations missing from reference list:

Aoki et al., 2003 and 2000

After English edition, Aoki et al., 2000 was not cited

Aoki, T., Hachikubo A., and Hori, M.: Effects of snow physical parameters on shortwave broadband albedos, J. Geophys. Res.,108 (D19), 4616, https://doi.org/10.1029/2003JD003506, 2003.

Tedesco and Kim 2006

Tedesco, M., and Kim, E.J.: Intercomparison of electromagnetic models for passive microwave remote sensing of snow, Ieee Transactions on Geoscience and Remote Sensing, 44, 2654-2666. https://doi.org/10.1109/TGRS.2006.873182, 2006

Royer et al. 2017

Royer, A., Roy, A., Montpetit, B., Saint-Jean-Rondeau, O., Picard, G., Brucker, L., and Langlois, A.: Comparison of commonly-used microwave radiative transfer models for snow remote sensing, Remote Sensing of Environment, 190, 247-259. https://doi.org/10.1016/j.rse.2016.12.020, 2017.

Zheng et al. 2021a and 2021b -> only Zheng et al. 2021 listed; however, there is a Zhang, Zheng et al. 2021.

One of Zheng et al., 2021 should be Zhang et al., 2021

Roesch et al. 1999

Roesch, A., Gilgen, H., Wild, M., and Ohmura, A.: Assessment of GCM simulated snow albedo using direct observations, Climate Dynamics, 15, 405– 418, https://doi.org/10.1007/s003820050290, 1999.

Mabuchi et al. 1997

Mabuchi, K., Sato, Y., Kida, H., Saigusa, N. and Oikawa, T.: A biosphere-atmosphere interaction model (BAIM) and its primary verification using grassland data, Papers in Meteorology and Geophysics, 47, 115– 140, https://doi.org/10.2467/mripapers.47.115, 1997.

In reference list but not cited in text:

Jordan, R.E.: A One-Dimensional Temperature Model for a Snow Cover: Technical Documentation for SNTHERM.89; U.S. Army Cold Regions Research and Engineering Laboratory: Hanover, NH, USA, 1991.

Yang, Z.L., Dickinson, R.E., Robock, A., and Vinnikov, K.Y. : Validation of the snow submodel of the biosphere-atmosphere transfer scheme with Russian snow cover and meteorological observational data. Journal of Climate, 10, 353-373, doi:

10.1175/1520-0442(1997)010<0353:Votsso>2.0.Co;2, 1997.

Zhang, P., Zheng, D., van der Velde, R., Wen, J., Zeng, Y., Wang, X., Wang, Z., Chen, J., and Su, Z.: Status of the Tibetan Plateau observatory (Tibet-Obs) and a 10-year (2009–2019) surface soil moisture dataset. Earth Syst. Sci. Data, 13, 3075–3102, https://doi.org/10.5194/essd-13-3075-2021, 2021.

Re: They were deleted, Zhang et al. 2021 remained.

Reference missing year:

Essery, R.: A factorial snowpack model (FSM 1.0). Geosci. Model Dev. 2015, 8, 3867–3876. [should be 2015]

Re: it was completed

Essery, R.: A factorial snowpack model (FSM 1.0), Geosci. Model Dev. 8, 3867–3876, https://doi.org/10.5194/gmd-8-3867-2015, 2015.

Duplicate references:

Roy, A., Picard, G., Royer, A., Montpetit, B., Dupont, F., Langlois, A., Derksen, C., and Champollion, N.: Brightness Temperature Simulations of the Canadian Seasonal Snowpack Driven by Measurements of the Snow Specific Surface Area. Ieee Transactions on Geoscience and Remote Sensing, 51, 4692-663 4704, doi: 10.1109/Tgrs.2012.2235842, 2013. 664

AND

Roy, A., Picard, G., Royer, A., Montpetit, B., Dupont, F., Langlois, A., Derksen, C., and Champollion, N.: Brightness Temperature Simulations of the Canadian Seasonal Snowpack Driven by Measurements of the Snow Specific Surface Area, IEEE T. Geosci. Remote, 51, 4692–4704, 667 doi:10.1109/TGRS.2012.2235842, 2013

Re: One of them remained and was modified to:

Roy, A., Picard, G., Royer, A., Montpetit, B., Dupont, F., Langlois, A., Derksen, C., and Champollion, N.: Brightness Temperature Simulations of the Canadian seasonal snowpack driven by measurements of the snow specific surface area, Ieee T. Geosci. Remote., 51, 4692-4704, https://doi.org/10.1109/Tgrs.2012.2235842, 2013.

---

## Author Response (AR3)

Dear Editor,

Thanks for all comments.

We have modified figures to make sure readers with color vision deficiencies to correctly interpret them. They are figure 1, figure 7, figure 9b, figure 12, figure 13 and figure 14.

Two papers introducing National Tibetan plateau data center were cited.

**Comments to the author:**

1. Since you have put the data at the National Tibetan Plateau/Third Pole Environment Data Center, you are welcome to cite the relevant introduction papers into the articles as: https://doi.org/10.1175/BAMS-D-21-0004.1 and https://doi.org/10.1175/BAMS-D-19-0280.1

Re: We cited these two papers:

Li, X., Che, T., Li, X. W., Wang, L., Duan, A. M., Shangguan, D. H., Pan, X. D., Fang, M. and Bao, Q.: CASEarth Poles Big Data for the Three Poles, B. Am. Meteorol. Soc., 101(9): E1475-E1491, https://doi.org/10.1175/Bams-D-19-0280.1, 2020.

Pan, X. D., Guo, X. J., Li, X., Niu, X. L., Yang, X. J., Feng, M., Che, T., Jin, R., Ran, Y. H., Guo, J. W., Hu, X. L. and Wu, A. D.: National Tibetan Plateau Data Center Promoting Earth System Science on the Third Pole, B. Am. Meteorol. Soc., 102(11), E2062-E78, https://doi.org/Bams-D-21-0004.1, 2021.

L81-83: Section 3 introduces the consolidated data that was released at the National Tibetan Plateau Data Center, China, which provides data support for international science programs (Li et al., 2021; Pan et al., 2021).

2.Please ensure that the colour schemes used in your maps and charts allow readers with colour vision deficiencies to correctly interpret your findings. Please check your figures using the Coblis – Color Blindness Simulator (https://www.color-blindness.com/coblis-color-blindness-simulator/) and revise the colour schemes accordingly.

Re: We put all images into the color blindness simulator to check the color scheme. To make sure readers with color vision deficiencies to correctly interpret the findings, we made the following revision:

Figure 1: in the original figure, we used different color to label different observation field. In this revised version, we used letter to label them instead of color. And corresponding texts were revised.

[Figure]

**Figure 1.** Location of the Altay National Reference Meteorological station (ANRMS) in Asia, along with the four test sites in the ANRMS. The black rectangle field (approximately 40 m × 50 m) was for snow layering, layer thickness, snow density, snow grain size and shape of each layer. The pink rectangle (approximately 60 m × 50 m) was for microwave radiometers observations. The blue rectangle field was for meteorological and soil data collection operated by the ANRMS. The red rectangle was for the automatically observation of the snow temperature, and 4-component radiation, designed by Northwest Institute of Eco-Environment and Resources, Chinese Academy of Science (NIEER).

Was revised to

[Figure]

**Figure 1. Location of the Altay National Reference Meteorological station (ANRMS) in Asia, along with the four test sites in the ANRMS. F1 (approximately 40 m × 50 m) was for snow layering, layer thickness, snow density, snow grain size and shape of each layer. F2 (approximately 60 m × 50 m) was for microwave radiometers observations. F3 was for meteorological and soil data collection operated by the ANRMS. F4 was for the automatically observation of the snow temperature, and 4-component radiation, designed by Northwest Institute of Eco-Environment and Resources, Chinese Academy of Science (NIEER).**

**Figure 7:**

[Figure]

[Figure]

**Figure 9b:**

[Figure]

[Figure]

**Figure 12:**

[Figure]

was revised to

[Figure]

**Figure 13:**

[Figure]

was revised to

[Figure]

**Figure 14:**

[Figure]

was revised to

[Figure]

**Figure 15:**

was revised to